# Systematics and Phylogenetic Interrelationships of the Enigmatic Late Jurassic Shark *Protospinax annectans* Woodward, 1918 with Comments on the Shark–Ray Sister Group Relationship

Patrick L. Jambura [1,2,*], Eduardo Villalobos-Segura [1], Julia Türtscher [1,2], Arnaud Begat [1,2], Manuel Andreas Staggl [1,2], Sebastian Stumpf [1], René Kindlimann [3], Stefanie Klug [4], Frederic Lacombat [5], Burkhard Pohl [5,6], John G. Maisey [7], Gavin J. P. Naylor [8] and Jürgen Kriwet [1,2]

1   Department of Palaeontology, University of Vienna, Josef-Holaubek-Platz 2, 1090 Vienna, Austria
2   Vienna Doctoral School of Ecology and Evolution (VDSEE), University of Vienna, Djerassiplatz 1, 1030 Vienna, Austria
3   Haimuseum und Sammlung R. Kindlimann, 8607 Aathal-Seegräben, Switzerland
4   School of Science (GAUSS), Georg–August Universität Göttingen, 37077 Göttingen, Germany
5   Interprospekt Group, 1724 Ferpicloz, Switzerland
6   Wyoming Dinosaur Center, Thermopolis, WY 82443, USA
7   Department of Vertebrate Paleontology, American Natural History Museum, New York, NY 10024, USA
8   Florida Museum of Natural History, University of Florida, Gainesville, FL 32611, USA
*   Correspondence: patrick.jambura@gmail.com

**Abstract:** The Late Jurassic elasmobranch *Protospinax annectans* is often regarded as a key species to our understanding of crown group elasmobranch interrelationships and the evolutionary history of this group. However, since its first description more than 100 years ago, its phylogenetic position within the Elasmobranchii (sharks and rays) has proven controversial, and a closer relationship between *Protospinax* and each of the posited superorders (Batomorphii, Squalomorphii, and Galeomorphii) has been proposed over the time. Here we revise this controversial taxon based on new holomorphic specimens from the Late Jurassic Konservat-Lagerstätte of the Solnhofen Archipelago in Bavaria (Germany) and review its skeletal morphology, systematics, and phylogenetic interrelationships. A data matrix with 224 morphological characters was compiled and analyzed under a molecular backbone constraint. Our results indicate a close relationship between *Protospinax*, angel sharks (Squatiniformes), and saw sharks (Pristiophoriformes). However, the revision of our morphological data matrix within a molecular framework highlights the lack of morphological characters defining certain groups, especially sharks of the order Squaliformes, hampering the phylogenetic resolution of *Protospinax annectans* with certainty. Furthermore, the monophyly of modern sharks retrieved by molecular studies is only weakly supported by morphological data, stressing the need for more characters to align morphological and molecular studies in the future.

**Keywords:** phylogenetics; elasmobranch evolution; calibration fossil; molecular backbone constraint; hypnosqualea; Mesozoic; Solnhofen Archipelago; Konservat-Lagerstätte

## 1. Introduction

Phylogenetic analyses are the foundation for many evolutionary studies, providing vital information about evolutionary rates, origination, diversification, and extinction of certain phylogenetic units of different ranks [1–3], and thus contribute to our understanding of the inherent drivers of biological diversity. Phylogenetics often play a pivotal role in other disciplines as well, such as epidemiology [4,5], climate change studies [6–9], or conservation biology [10–13]. For many of these studies it is essential to calibrate the phylogenetic tree, which is usually performed with the help of fossil taxa (i.e., fossil calibration) [14,15].

Unfortunately, most fossils are fragmentary, hampering our ability to incorporate them confidently within a phylogenetic hypothesis.

The fossil record of sharks and rays (Elasmobranchii) mainly consists of isolated teeth, which are rapidly grown and continuously replaced. By contrast, skeletal remains are rare due to the poor preservation potential of the cartilaginous endoskeleton. However, a few localities, so-called Konservat-Lagerstätten, are known to harbor skeletal material with exquisite preservation (sometimes even with soft tissue preservation) and, therefore, offer a unique window into the past. The main Konservat-Lagerstätten for fossil elasmobranchs are several localities in Bavaria, which are collectively referred to as the Solnhofen Archipelago (Germany; Late Jurassic) [16,17], Nusplingen (Germany; Late Jurassic) [18], Cerin (France; Late Jurassic) [19–21], Hakel, Hadjoula, Sahel Alma, and Nemoura (Lebanon; Late Cretaceous) [22–24], Monte Bolca (Italy; Eocene) [25–27], the Green River Formation (Wyoming; Early Eocene) [28], Rauenberg "Frauenweiler" (Germany; Oligocene) [29,30], Froidefontaine (France; Oligocene) [31–33], and the Pisco Formation (Peru; Miocene–Pleistocene) [34–36].

Elasmobranchii (sharks and rays) comprises three clades ("superorders"): Batomorphii (rays), Galeomorphii ("galeomorph sharks"), and Squalomorphii ("squalomorph sharks") [37]. Although these groups are well-supported by morphological and molecular phylogenies, there are conflicting views on the interrelationships of these groups between both approaches [38]. Whereas molecular data recognize an adelphotaxa (sister group) relationship between sharks (Galeomorphii + Squalomorphii) and rays (Batomorphii) [37,39–43], morphological data generally retrieve a paraphyletic shark group, in which batomorphs are resolved as highly derived (squalomorph) sharks [44–48]. The morphological hypothesis (i.e., "hypnosqualean hypothesis" of Shirai [44,45]) suggests the presence of a highly derived squalomorph clade, formerly classified as the Superorder Hypnosqualea [47], which comprises angel sharks (Squatiniformes), saw sharks (Pristiophoriformes), and rays (Batomorphii). The fossil record suggests that the three modern superorders probably did not originate later than the Early Jurassic [38,49–51]. However, Pristiophoriformes and Squaliformes, which lie at a basal position compared with Batomorphii in morphological phylogenies [45,46,48], are not known to occur before the Cretaceous [38,50]. Thus, the fossil evidence favors the sister relationship proposed by molecular data over the Hypnosqualea hypothesis.

A taxon that has historically played a significant role in discussions surrounding elasmobranch interrelationships is the Jurassic genus *Protospinax* Woodward, 1918. Teeth of this genus have been recovered from a number of Jurassic localities throughout Europe and Russia [52], but it is best known by the type species *Protospinax annectans* Woodward, 1918, which is known from a number of well-preserved holomorphic specimens from the Solnhofen Archipelago [47,53,54]. Despite the excellent preservation of these fossils, its phylogenetic relations with other elasmobranchs have been an enigma since the first description of this species more than 100 years ago. Woodward [53] suggested a close affiliation with squaliform and echinorhiniform sharks (i.e., "Spinacidae") within a clade that also included batomorphs ("Tectospondyli"). Since that time, *Protospinax* has been hypothesized to represent either the ancestor of a clade comprising batomorphs, several squalomorph and galeomorph sharks [55], the base of Squalomorphii and Batomorphii [20], an intermediate group between Pristiophoriformes and batomorphs [56], a stem squalomorph [57], a sister group to squaliform sharks [46], or a squaliform shark [58,59]. All these hypotheses were based on the type of material (holo- and paratype). Maisey [54] described a new specimen (SNSB-BSPG 1963-1-19) and revised the type material. He concluded that the new specimen and the holotype belonged to the Batomorphii and were closely allied with the Late Jurassic batomorph *Belemnobatis sismondae*, leading to their assignment to *Belemnobatis annectans*, whereas the paratype was described as a new galeomorph shark species, *Squalogaleus woodwardi* [54]. In a later study, de Carvalho and Maisey [47] refuted this interpretation and synonymized *Belemnobatis annectans* and *Squalogaleus woodwardi* with *Protospinax annectans* and regarded *Protospinax* as a squalomorph shark.

Even though *Protospinax annectans* is represented by several well-preserved holomorphic specimens, it has rarely been included in cladistic analyses. De Carvalho and Maisey [47] were the first to conduct a phylogenetic analysis based on dental and skeletal characters following cladistic principles and hypothesized that *Protospinax annectans* was a stem group hypnosqualean closely related to a clade comprising Squatiniformes, Pristiophoriformes, and Batomorphii. Support for this hypothesis came from subsequent phylogenetic studies based on dental characters, which also proposed a close relationship between *Protospinax*, Squatiniformes, and Pristiophoriformes [60,61], although a closer relationship between *Protospinax* and Pristiophoriformes was suggested by Adnet and Cappetta [60]. Conversely, a recent morphological study based on new data proposed a closer relationship between *Protospinax* and Squaliformes [59]. Despite the questionable phylogenetic position of *Protospinax*, it has nevertheless been attributed an important place in the evolutionary history of elasmobranchs [54] and despite the uncertainty surrounding its phylogenetic relationships, it is used as an important fossil for calibrating molecular phylogenies [13,61,62].

Here, we present new holomorphic specimens of *Protospinax annectans* and revise the skeletal morphology of this taxon within a molecular framework by applying a molecular backbone constraint. Molecular backbone constraints have been used previously to incorporate fossils within a molecular hypothesis in a number of different groups (e.g., snakes [63], turtles [64], old world monkeys [65], and leafy liverwort [66]), but this has never been performed before on cartilaginous fish. By applying this method, we aim to identify unambiguous synapomorphies within the Elasmobranchii, discuss previously proposed morphological characters and character evolution in the light of molecular evidence, and try to decipher the phylogenetic position of the enigmatic Late Jurassic elasmobranch *Protospinax annectans*.

## 2. Materials and Methods

### 2.1. Material

The fossil specimen (PBP-SOL-8007) that forms the focus of this study is preserved on a single slab of lithographic limestone measuring 125 cm in maximum height, 60 cm in maximum width, and 3.5 cm in thickness. For comparison and to complement morphological information, nine additional specimens (four of which were previously unknown to the scientific literature (Figure 1)) were studied (see Section 2.3). All specimens came from the early Tithonian Altmühltal Formation of Solnhofen and Eichstätt in Bavaria, Germany [67–69], which is known for its diverse, rich, and exquisitely preserved fossil content, including articulated skeletal remains of Chondrichthyes (e.g., [16,17,70]).

Specimen PBP-SOL-8007 was photographed using a Nikon D5300 DSLR camera with a mounted AF-S DX Micro NIKKOR 40 mm f/2.8G lens. Close-up photographs of the teeth and scales were taken with a Nikon D7500 DSLR camera with a mounted Laowa 25 mm 2.8 2.5-5x Ultra-Macro lens or a mounted Plan 10 microscope lens. In addition, photographs were taken under ultraviolet (UV) light following the technique described in Tischlinger and Arratia [71] with a Raytech 10-075 LS-4CB UV Long/Short-Wave Lamp (230 Volts/50 Hz, 4 Watts).

The tooth histology of *Protospinax annectans* was examined non-destructively with X-ray-computed tomography (CT) imaging using a Xradia MicroXCT system (Zeiss, Oberkochen, Germany) at the Department of Theoretical Biology (University of Vienna, Austria). The examined tooth (EMRG-Chond-T-80) was from the Solnhofen Archipelago but was found isolated and was not associated with any holomorphic specimen. The scan was performed at 1.9 µm resolution with 40 kV source voltage and 125 µA source current. The resulting stack file was loaded into the software system Amira (version 5.4.5, FEI Visualization Sciences Group, Hillsboro, OR, USA) and virtual sections were created with the ObliqueSlice command. Images were edited regarding color balance and contrast in Adobe Photoshop CS6 (version 13.0, Adobe Systems, San José, CA, USA).

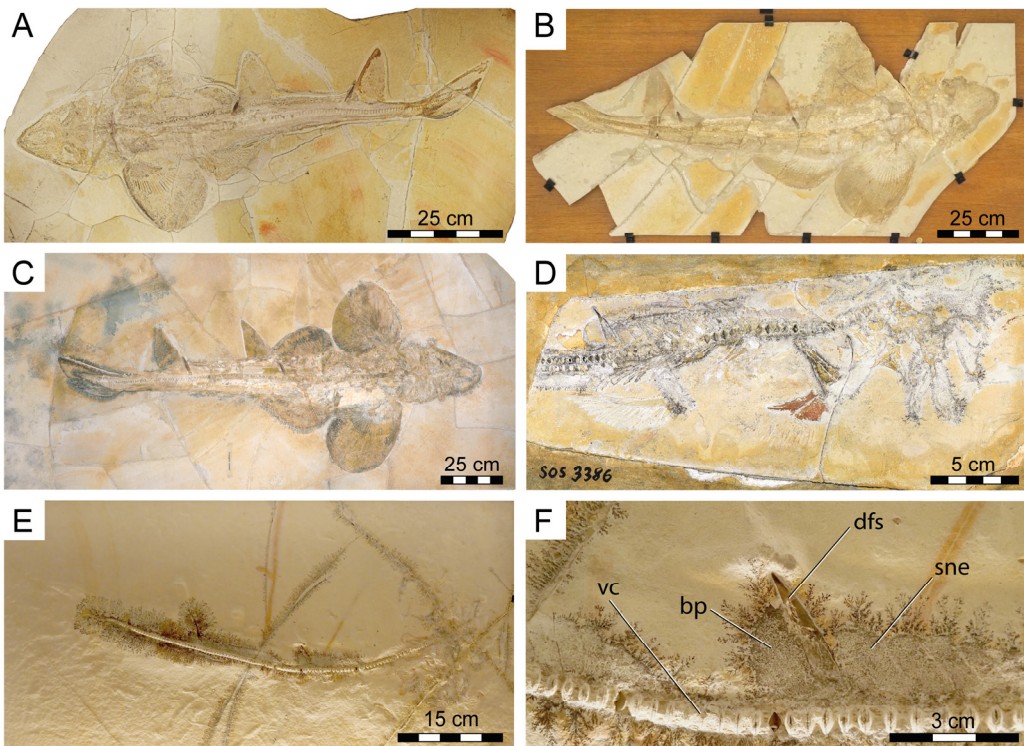

**Figure 1.** New fossil skeletal material of *Protospinax annectans* Woodward, 1918 examined in this study. (**A**) PBP-SOL-8007; (**B**) MB 14-12-22-1; (**C**) UMN uncatalogued; (**D**) JME-SOS 3386; (**E**,**F**) FSM 727. *Abbreviations*: **bp**, basal plate; **dfs**, dorsal fin spine; **sne**, supraneuralia; **vc**, vertebral column.

Measurements of the body proportions followed the scheme proposed by Compagno [72] (Figure 2) and were taken digitally using the freeware tool Fiji [73]. Measurements of the vertebral centra were taken manually with a fiberglass slide caliper providing an accuracy of 0.1 mm (Table S1).

*2.2. Phylogenetic Analyses*

For the phylogenetic analyses, a matrix containing 52 taxa (43 extant and nine fossil taxa) and 224 morphological characters was compiled using Mesquite 3.70 [74] (Supplementary Material S1). The ingroup included 17 squalomorph sharks (15 extant taxa and two fossil taxa), 12 galeomorph sharks (all 12 were extant taxa) and 17 batomorphs (14 extant taxa and three fossil taxa) of all 13 extant orders: Carcharhiniformes (n = 6), Echinorhiniformes (n = 1), Heterodontiformes (n = 1), Hexanchiformes (n = 4), Lamniformes (n = 3), Orectolobiformes (n = 2), Pristiophoriformes (n = 2), Squaliformes (n = 7), Squatiniformes (n = 2), Myliobatiformes (n = 4), Rajiformes (n = 2), Rhinopristiformes (n = 4), and Torpediniformes (n = 4). Six taxa were chosen as outgroup taxa: one Symmoriiformes from the Late Carboniferous (*Cobelodus aculeatus*), two extant Holocephali (*Chimaera*, *Harriotta*), and three Paleozoic and Mesozoic Hybodontiformes (*Egertonodus*, *Hamiltonichthys*, *Hybodus*). Outgroup selection is critical for determining the polarity of characters and the use of multiple outgroups has been demonstrated to increase phylogenetic accuracy within the ingroup [75,76].

Characters were mainly based on Landemaine et al. [77] and Villalobos-Segura et al. [59], whose matrices were largely assembled from previous phylogenetic studies on sharks [46–48,78] and rays [79–95], most of which were modified versions of previous studies and can be traced back to Shirai [45] for sharks and McEachran et al. [96] for rays. Additionally, dental characters, which have been used for phylogenetic analyses in squalomorph sharks [60,61,97] and skeletal characters defining galeomorph sharks [98–100] and outgroup taxa [101,102] were included. Eight new characters were added: char. 57 sustentaculum

(following observations of Motta and Wilga [103]); char. 103 number of facets on the scapulocoracoid (following observations of da Silva and de Carvalho [104]); char. 141 number of vertebrae with expanded basiventrals (if expanded basiventrals are present); char. 147 pleural ribs (following observations of Maisey [105]); char. 155 number of basal plates (if the endoskeleton is composed of triangular or rectangular basal cartilage); char. 178 number of fin spines (if dorsal fin spines present); char. 193 tooth crown dentine (following observations of Jambura et al. [106–108]); and char. 194 pulp cavity (following observations of Jambura et al. [106–108]). For a detailed character list see Supplementary Material S2. Previously established characters and character states were reviewed using extant and fossil material from different collections (see the subsequent Sections 2.3 and 2.4). In cases where no direct examination of the material was possible, published images and illustrations were used to review morphological characters [45,80,98,104,109–144]. If characters could not be reviewed in either the available material, or on the images, the original coding was retained. Characters were coded following the procedure for reductive coding proposed by Brazeau [145] in order to avoid the lumping of hierarchically different states (i.e., presence/absence of a structure and different states of presence in one character; "no tail; red tail; blue tail" problem [146]).

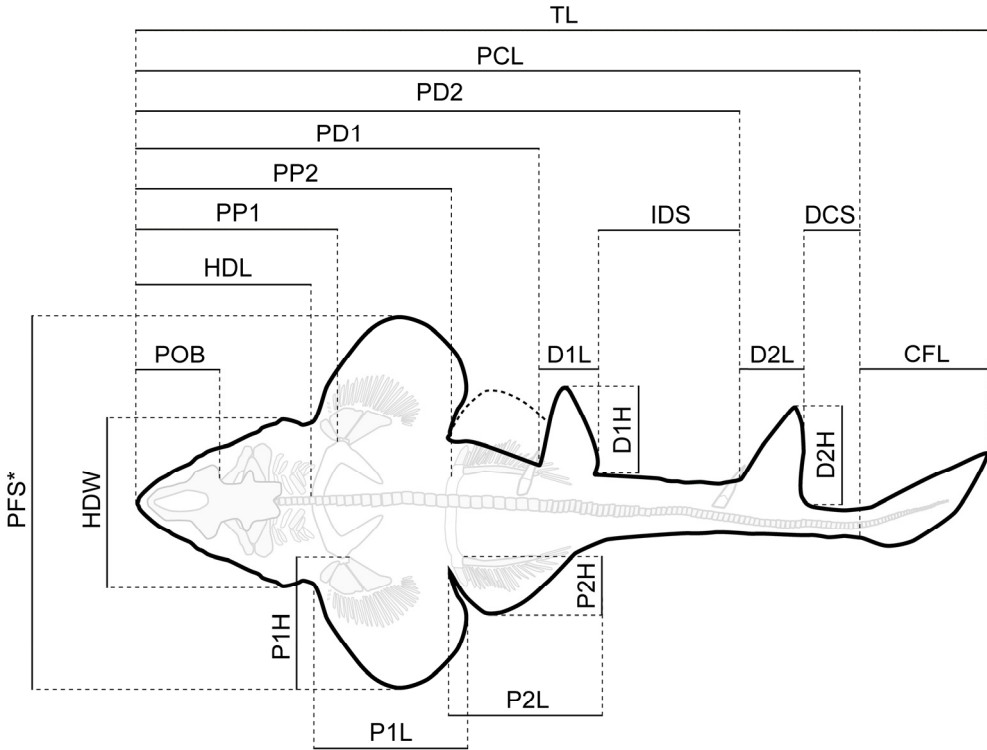

**Figure 2.** Simplified outline drawing of *Protospinax annectans* Woodward, 1918 from the lower Tithonian Solnhofen Archipelago showing the measurements listed in Table 1. *Abbreviations*: **CFL**, caudal fin length; **D1H**, first dorsal height; **D2H**, second dorsal height; **D1L**, first dorsal length; **D2L**, second dorsal length; **DCS**, dorsal caudal space; **HDL**, head length; **HDW**, head width; **IDS**, interdorsal space; **P1H**, pectoral height; **P2H**, pelvic height; **P1L**, pectoral inner margin length; **P2L**, pelvic inner margin length; **PCL**, precaudal length; **PD1**, pre-first dorsal length; **PD2**, pre-second dorsal length; **PFS**, pectoral fin span; **POB**, preorbital length; **PP1**, prepectoral length; **PP2**, prepelvic length; **TL**, total length. *The pectoral fin of the right side was partly missing and PFS was estimated by measuring the distance between the tip of the pectoral fin and the vertebral column (perpendicular to the axial axis) and then multiplying this distance by two.

**Table 1.** Morphometric data for three well-preserved specimens (PBP-SOL-8007, MB 14-12-22-1, UMN uncatalogued) and the holotype (NHMUK PV P 8775) of *Protospinax annectans* Woodward, 1918 from the lower Tithonian of the Solnhofen Archipelago, Bavaria, Germany. Measurements followed Compagno [72] and are summarized in Figure 2. *Abbreviations*: **CFL**, caudal fin length; **D1H**, first dorsal height; **D2H**, second dorsal height; **D1L**, first dorsal length; **D2L**, second dorsal length; **DCS**, dorsal caudal space; **HDL**, head length; **HDW**, head width; **IDS**, interdorsal space; **P1H**, pectoral height; **P2H**, pelvic height; **P1L**, pectoral inner margin length; **P2L**, pelvic inner margin length; **PCL**, precaudal length; **PD1**, pre-first dorsal length; **PD2**, pre-second dorsal length; **PFS**, pectoral fin span; **POB**, preorbital length; **PP1**, prepectoral length; **PP2**, prepelvic length; **TL**, total length.

| | PBP-SOL-8007 | | MB 14-12-22-1 | | UMN Uncatalogued | | NHMUK PV P 8775 * | |
| **Measurements** | cm | % of PCL | cm | % of PCL | cm | % of PCL | cm | % of PCL |
|---|---|---|---|---|---|---|---|---|
| TL | 103.2 | 116.96 | - | - | 163.3 | 118.33 | - | - |
| PCL | 88.2 | 100.00 | 119.1 | 100.00 | 138.0 | 100.00 | 84.6 | 100 |
| POB | 9.8 | 11.05 | 13.4 | 11.21 | 15.0 | 10.86 | 10.6 | 12.54 |
| HDL | 22.3 | 25.31 | 30.4 | 25.55 | 33.9 | 24.58 | 27.5 | 32.6 |
| HDW | 17.0 | 19.29 | 20.7 | 17.37 | 22.9 | 16.56 | 19.9 | 23.5 |
| PP1 | 25.9 | 29.38 | 33.2 | 27.91 | 36.4 | 26.38 | - | - |
| P1L | 18.5 | 20.93 | 26.8 | 22.47 | 28.8 | 20.85 | 18.7 | 22.09 |
| P1H | 17.9 | 20.26 | 22.2 | 18.66 | 27.2 | 19.74 | 13.6 | 16.02 |
| PFS | 46.1 | 52.25 | 59.6 | 50.04 | 74.2 | 53.79 | 41.8 | 49.4 |
| PD1 | 48.9 | 55.40 | 63.6 | 53.39 | 77.6 | 56.21 | - | - |
| D1L | 8.4 | 9.48 | 12.5 | 10.45 | 12.0 | 8.73 | - | - |
| D1H | 12.5 | 14.18 | 13.6 | 11.40 | 15.2 | 10.98 | - | - |
| IDS | 15.8 | 17.95 | 23.6 | 19.85 | 24.4 | 17.65 | - | - |
| PD2 | 73.2 | 82.97 | 98.6 | 82.82 | 114.3 | 82.85 | - | - |
| D2L | 8.1 | 9.18 | 10.7 | 9.01 | 11.5 | 8.36 | - | - |
| D2H | 11.5 | 12.99 | 15.2 | 12.74 | 17.4 | 12.64 | - | - |
| DCS | 6.7 | 7.60 | 9.1 | 7.67 | 13.5 | 9.78 | - | - |
| PP2 | 40.1 | 45.51 | 50.4 | 42.27 | 56.7 | 41.08 | 41.4 | 48.96 |
| P2L | 19.0 | 21.57 | 28.6 | 24.04 | 29.5 | 21.37 | 15.7 | 18.61 |
| P2H | 7.3 | 8.21 | 11.5 | 9.64 | 10.5 | 7.63 | 7.5 | 8.91 |
| CFL | 15.0 | 16.96 | - | - | 25.0 | 18.13 | - | - |

* The holotype has been recovered in five separate pieces and thus, PCL and all derived relative values are estimates.

A parsimony analysis was executed in the command-line version of TNT 1.5 for MacOS [147]. For the phylogenetic analysis we applied a molecular backbone constraint on the extant taxa following the topology of Stein et al. [13]. For this purpose, a phylogeny subset of 10,000 trees was generated from the VertLife website (https://vertlife.org/phylosubsets/ accessed on 25 November 2022) to match the reduced number of taxa in our matrix. The subset was then loaded into TreeAnnotator in Beast 2.7.1 to create a maximum credibility tree, which served as the backbone tree (Figure S1). The default settings were used in TreeAnnotator (burn in percentage 10; posterior probability limit 0.0), for the node heights, median heights instead of common ancestor heights were applied. Only nodes that were well-supported by molecular data were enforced, whereas all nodes with a posterior probability of less than 70% were collapsed in the molecular backbone constraint tree and thus were allowed to be overturned by morphological data in our analysis (see Lee [63]). The fossil taxa remained unconstrained and were thus allowed to fall anywhere within the constrained topology of the extant taxa. An unconstrained tree and a tree with fully enforced constraints were also computed and are provided in the Supplementary Material (Figures S2 and S3). In TNT, traditional search methods were used with the following parameters: random sequence addition, using tree bisection and reconnection (TBR) algorithm for branch permutations with 1000 iterations, holding ten trees for each iteration.

Additionally, unconstrained parsimony and maximum likelihood (ML) analyses were conducted in PAUP V4 [148] and compared with the results generated by TNT (Figure S2). For the parsimony analysis in PAUP, a heuristic search with 1000 replicates of random stepwise addition (branch swapping: TBR) holding one tree at each step was conducted.

The ML analysis was conducted employing the Mkv evolutionary model (Markov K model for discrete morphological data with only variable characters) [149]. A neighbor-joining tree was used as a starting tree. For the subsequent tree search, the TBR algorithm was used for branch permutations with 10,000 replications, assuming a gamma distribution across characters. Protocols and scripts for all phylogenetic analyses are provided in the Supplementary Material (Supplementary Material S3, Material S4, Material S5, Material S6, Material S7, Material S8, Material S9).

*2.3. Reviewed Fossil Material*

*Belemnobatis sismondae* (CM 4044, very incomplete specimen, only pectoral and pelvic fin preserved; CM 4070, incomplete specimen exposed in dorsal view, exhibiting only the pectoral and pelvic girdle and part of the vertebral column; CM 5131, incomplete specimen preserved in dorsal view, missing the rostrum, parts of the right pectoral and pelvic fin and the caudal region; MNHN CRN 13, incomplete specimen in ventral view, missing the caudal region and the right pectoral and pelvic fins; and MNHN CRN 68, incomplete specimen, exhibiting part of the pectoral fin and pelvic fin); *Kimmerobatis etchesi* (MJML K874, holotype, almost complete specimen in dorsal view; MJML K1894, paratype, pelvic region, claspers, and parts of the caudal region preserved; both specimens figured in Underwood and Claeson [94]); *Protospinax annectans* (FSM 727a,b, part and counterpart of a disarticulated specimen preserving the cranium, vertebral column, dorsal fin spines, and supraneuralia; JME-SOS 3386, partly preserved specimen; MB 14-12-22-1, complete specimen with claspers, exposed in dorsal view; MCZ VPF-278, complete specimen in dorsal view; MCZ VPF-6394a,b part and counterpart of a nearly complete disarticulated specimen; NHMUK PV P 8775a,b,c holotype, part and counterpart of a nearly complete specimen in dorsal view; NHMUK PV P 37014, paratype, neurocranium, partly preserved dentition, vertebral column and two dorsal fin spines; SNSB-BSPG 1963-I-19, complete specimen in ventral view; PBP-SOL-8007, complete specimen in dorsal view; and UMN uncatalogued, complete specimen in dorsal view); *Pseudorhina alifera* (CM 4052, complete specimen in dorsal view; CM 4054, complete specimen in dorsal view; CM 5397, complete specimen in ventral view; NHMUK PV P 8535, complete specimen in dorsal view; NHMUK PV P 37013, complete specimen in dorsal view; NHMUK PV P 37370, cast of type, complete specimen in dorsal view; NHMUK PV P 37997, cast, complete specimen in dorsal view; NHMUK PV P 49149, cast of type, complete specimen in dorsal view; and USNM V 21056, part and counter part of a complete specimen in dorsal view); *Pseudorhina acanthoderma* (NHMUK PV P 8784, complete specimen, exposed in dorsal view; and NHMUK PV P 38002, incomplete specimen, lacking parts of the neurocranium and the caudal region, exposed in dorsal view); and *Spathobatis bugesiacus* (CM 4408, complete specimen in dorsal view; CM 5396, complete specimen in ventral view; NHMUK PV P 2099, incomplete specimen lacking rostrum and caudal region, exposed in ventral view; NHMUK PV P 10934, complete specimen in ventral view; and SNSB-BSPG 1952-I-82, whole specimen in dorsal view).

*2.4. Reviewed Extant Material*

*Carcharhinus galapagensis* (GMBL uncatalogued, CT scan); *Carcharhinus plumbeus* (GMBL 79-60, CT scan); *Carcharodon carcharias* (MCZ 171013, CT scan); *Chimaera cubana* (USNM 400700, CT scan); *Chlamydoselachus anguineus* (UF 44302, CT scan); *Dalatias licha* (AMNH Mauritius 4582, CT scan); *Deania calceus* (AMNH MB85-015114, CT scan); *Etmopterus splendidus* (AMNH 258170, CT scan); *Ginglymostoma cirratum* (USNM 127110, CT scan); *Gymnura altavela* (GMBL 81-86, CT scan); *Harriotta raleighana* (USNM 320579, CT scan); *Hemipristis elongata* (LACM 37712-1, CT scan); *Hemiscyllium ocellatum* (AMNH 44128, CT scan); *Heptranchias perlo* (GMBL 96-12, CT scan); *Heterodontus francisci* (AMNH 96795, CT scan); *Hexanchus nakamurai* (UF 165855, CT scan); *Hypnos monopterygius* (USNM 84374, CT scan); *Isurus oxyrinchus* (GMBL 8446, CT scan); *Mobula munkiana* (SIO 85-34, CT scan); *Mollisquama mississippiensis* (TU 203676, CT scan. Additionally, PB-CT

scans of the cranium of the specimen are figured in Denton et al. 2018, figs 1–12 [132]); *Mustelus manazo* (AMNH 258162, CT scan); *Notorynchus cepedianus* (UCMP uncatalogued, CT scan of a neurocranium [130]); *Oxynotus centrina* (USNM 206065, CT scan); *Platyrhinoidis triseriata* (LACM 32, CT scan); *Pristiophorus nudipinnis* (CSIRO uncatalogued, CT scan); *Pseudocarcharias kamoharai* (LACM 45857, CT scan); *Raja eglanteria* (GMBL 02-155, also GMBL 5557, CT scan); *Rhina ancylostoma* (LACM 38117-38, CT scan); *Rhinobatos lentiginosus* (GMBL 74-37, CT scan); *Rhinoptera bonasus* (GMBL 73-7, CT scan); *Rhizoprionodon terraenovae* (GMBL uncatalogued, CT scan); *Rhynchobatus springeri* (AMNH 258310, CT scan); *Scyliorhinus meadi* (GMBL 8312, CT scan); *Sphyrna lewini* (USNM 203101, CT scan); *Sphyrna media* (USNM 205377, CT scan); *Sphyrna tiburo* (AMNH uncatalogued, CT scan); *Sphyrna tudes* (USNM 159197, CT scan); *Sphyrna zygaena* (USNM 325631, CT scan); *Squalus acanthias* (GMBL 7313, CT scan); *Squalus brevirostris* (AMNH 258171, CT scan); *Squatina nebulosa* (AMNH 258172, CT scan); *Torpedo fuscomaculata* (USNM 320677, CT scan); and *Zameus squamulosus* (USNM 400734, CT scan).

All CT scans are whole-body scans and are available at https://sharksrays.org (accessed on 25 November 2022) (if not indicated otherwise).

*2.5. Institutional Abbreviations*

American Museum of Natural History (**AMNH**), New York, NY, USA; Carnegie Museum of Natural History (**CM**), Pittsburgh, USA; Commonwealth Scientific and Industrial Research Organisation (**CSIRO**), Canberra, Australia; Department of Palaeontology-University of Vienna (**EMRG**), Vienna, Austria; Fossilien und Steindruck Museum (**FSM**) (formerly Mayberg Museum), Gunzenhausen, Germany; Grice Marine Biological Laboratory-College of Charleston (**GMBL**), Charleston, SC, USA; Jura-Museum Eichstätt (**JME-SOS**), Eichstätt, Germany; Natural History Museum of Los Angeles County (**LACM**), Los Angeles, CA, USA; Museum Bergér (**MB**), Eichstätt, Germany; Museum of Comparative Zoology of Harvard University (**MCZ**), Cambridge, MA, USA; Museum of Jurassic Marine Life (**MJML**), Kimmeridge, UK; Muséum National d'Histoire Naturelle (**MNHN**), Paris, France; Natural History Museum (**NHMUK**), London, UK; Wyoming Dinosaur Center (**PBP-SOL**), Thermopolis, WY, USA; Scripps Institution of Oceanography (**SIO**), San Diego, CA, USA; Bayerische Staatssammlung für Paläontologie und Geologie (**SNSB-BSPG**), Munich, Germany; Tulane University Biodiversity Research Institute (**TU**), Belle Chasse, LA, USA; University of California Museum of Paleontology (**UCMP**), Berkeley, CA, USA; University of Florida (**UF**), Gainesville, FL, USA; Urweltmuseum Neiderhell (**UMN**), Raubling, Germany; and Smithsonian National Museum of Natural History (**USNM**), Washington, DC, USA.

## 3. Results

### 3.1. Systematic Paleontology

The terminology used here to describe the skeletal morphology mainly follows Compagno [150] and Shirai [45]. For the clasper and caudal fin skeleton more specifically, the terminologies proposed by Moreira and Carvalho [135] and Moreira et al. [151] were followed. Dental description follows the terminology of Cappetta [52]. Isolated teeth of *Protospinax annectans* were reported from a wide stratigraphic range, spanning from the Callovian (Middle Jurassic) [58] to the Aptian (Early Cretaceous) [152]. It is unclear whether all these reports resemble the same species, which was originally described based on skeletal material from the early Tithonian of Solnhofen. Revising all potential records of *Protospinax annectans* was beyond the scope of this manuscript, which is why we restricted the synonymy list to those based on articulated skeletal material only.

Class **Chondrichthyes** Huxley, 1880 [153];
Subclass **Elasmobranchii** Bonaparte, 1838, sensu Maisey, 2012 [154];
Superorder **Squalomorphii** Compagno, 1973 [56];
Order *incertae sedis*;
Family **Protospinacidae** Woodward, 1918 [53].

**Type genus**: *Protospinax* Woodward, 1918 [53];
**Included genera**: *Protospinax* Woodward, 1918 (monogeneric family).

Genus ***Protospinax*** Woodward, 1918 [53]

**Type species**: *Protospinax annectans* Woodward, 1918; lower Tithonian (Late Jurassic) of Solnhofen and Eichstätt, Germany.

**Included species**: *Protospinax annectans* Woodward, 1918 [53]; *Protospinax bilobatus* Underwood and Ward, 2004 [155]*; *Protospinax carvalhoi* Underwood and Ward, 2004 [155]*; *Protospinax lochensteinensis* Thies, 1983 [58]*; *Protospinax magnus* Underwood and Ward, 2004 [155]*; *Protospinax planus* Underwood, 2002 [156]* (asterisk denotes species based on isolated teeth only).

Another species, *Protospinax muftius* Thies 1983 [58]*, was described from the Callovian of South England. However, there is considerable doubt about the correct assignment to the genus *Protospinax* and it was suggested that *Protospinax muftius* was instead an orectolobiform shark [152,157,158]. We follow this assumption here and do not consider this species to belong to the genus *Protospinax*.

**Temporal and Spatial Distribution of the genus:** Early Jurassic to Early Cretaceous of Europe and Russia. Toarcian of Belgium [159] and Germany [160]; Bajocian of England [161]; Bathonian of England [155,158] and Poland [162]; Callovian of England [58], Poland [163], and Russia [164]; Callovian–Oxfordian of Poland [165]; Oxfordian of Germany [58,166,167]; Oxfordian–Kimmeridgian of Spain [168]; Kimmeridgian of England [156], France [167,169], Germany [167], and Spain [170]; Tithonian of Germany [47,53,54]; and Aptian of England [152].

***Protospinax annectans* Woodward, 1918**
1919  *Protospinax annectans* Woodward [53], pl. 1, figs. 2, 2a, 3, 3a;
1937  *Protospinax annectans* Woodward, 1918–White [55], text-fig. 8;
1949  *Protospinax annectans* Woodward, 1918–de Saint-Seine [20], text-fig. 10B;
1967  *Protospinax annectans* Woodward, 1919–Schaeffer [57], text: text-fig. 1–10A;
1976  *Belemnobatis annectans* (Woodward, 1919)–Maisey [54], pl. 111, text-figs. 1–4;
1976  *Squalogaleus woodwardi* Maisey [54], pl. 112, text-figs. 5–8;
1978  *Protospinax annectans* Woodward–Barthel [171], pl. 27;
1987  *Squalogaleus woodwardi* Maisey, 1976–Cappetta [172], text-fig. 55;
1987  *Protospinax annectans* Woodward, 1919–Cappetta [172], text-fig. 62;
1988  *Protospinax*-Carroll [173], text-figs. 5–16a;
1994  *Protospinax annectans* Woodward, 1890–Frickhinger [174], text-fig. 418;
1996  *Protospinax annectans* Woodward, 1919–de Carvalho and Maisey [47], text-figs. 3, 5–9;
1999  *Protospinax annectans* (Woodward, 1919)–Frickhinger [175], text-figs. 163, 164;
1999  *Protospinax* sp.Frickhinger [175], text-figs. 165, 166;
2004  *Protospinax annectans* Woodward, 1919–Kriwet and Klug [16], text-fig. 5;
2011  *Protospinax annectans* Woodward, 1919–Leidner and Thies [176], pl. 3;
2011  *Squalogaleus woodwardi* Maisey, 1976 (=*Protospinax annectans* Woodward 1919)–Leidner and Thies [176], pl. 4, 5;
2012  *Protospinax annectans* Woodward, 1918–Cappetta [52], text-figs. 135, 136;
2015  *Protospinax annectans* Woodward, 1919–Kriwet and Klug [17], text-figs. 700, 701.

**Holotype**: The type specimen of *Protospinax annectans* comprises an almost complete fossil skeleton from the early Tithonian (Late Jurassic) Altmühltal Formation of Solnhofen, Bavaria, Germany and is housed in the fossil fish collection of the Natural History Museum in London under collection number NHMUK PV P 8775a,b,c (formerly BM(NH) P.8775).

**Temporal and Spatial Distribution of the species:** Middle Jurassic to Early Cretaceous of Europe. Callovian of England [58]; Oxfordian of Germany [166]; Kimmeridgian of Germany [167]; Kimmeridgian–Tithonian of France [167]; Tithonian of Southern Germany [47,53,54]; Aptian of England [152];

**Referred specimens:** See Section 2.3 in the Materials and Methods section.

**Locality and age**: All articulated skeletons of *Protospinax annectans* are known from the early Tithonian (Late Jurassic) Altmühltal Formation of Solnhofen and Eichstätt, Bavaria, Germany.

### 3.1.1. Nomenclatural Notes

Some confusion exists about the year when *Protospinax annectans* was described by Woodward, with most authors either referencing 1918 or 1919. Sometimes even both years can be found within the same publication (e.g., [55,176,177]), or that the year is denoted with a question mark [160,167], further adding to the confusion about the true date of the erection of this species. *Protospinax annectans* was first mentioned during a scientific meeting of the Zoological Society of London in 1918. The corresponding abstract, however, was published in parts 3 and 4 of the 1918 volume of the "Proceedings of the Zoological Society of London", which was printed in March 1919 [53]. Nonetheless, according to the Zoological Record and its printed version of 1918 [178] (which was also compiled by the Zoological Society of London), *Protospinax annectans,* was erected in 1918. This interpretation was also followed by the work "The Genera of Fishes: A Contribution to the Stability of Scientific Nomenclature", authored by Jordan and Evermann [179]. The reason for the current discrepancy lies in the format of the journal "Proceedings of the Zoological Society of London", which bound the abstracts of the society's scientific meetings at the end of the volume, which was often published not earlier than the following year. However, those abstracts were published on the Tuesday following the meeting at which the paper was read [180] and, therefore, a printed version of Woodward's first description of *Protospinax* was already available in 1918 (the paper was read on June 11, 1918). Therefore, article 21.8.1 of the International Code of Zoological Nomenclature (ICZN) [181] comes into effect ("Before 2000, an author who distributed separates in advance of the specified date of publication of the work in which the material was published thereby advanced the date of publication") and 1918 should be regarded as the year of the first description of *Protospinax annectans*.

### 3.1.2. Revised Diagnosis

Medium-sized squalomorph shark, reaching a total length of approximately 1.7 m; crushing dentition; teeth small and arranged in diagonal rows; tooth crowns with flat occlusal surface, with or without a moderately cuspidate occlusal crest; labial apron absent; lingual uvula present; uvula flanked by one or two pairs of foramina; teeth with a hollow pulp cavity and orthodentine constituting the crown and root; head with short and rounded snout, detached from the pectoral fins; rostrum dorsally open, forming a lateroventrally walled scoop with a long prefrontal fontanelle; rostral appendices absent; subnasal fenestra absent; pronounced postorbital processes; orbital articulation present; basihyal slightly wider than deep; no synarcual present; large pectoral fins; scapulocoracoid with two condyles, one articulating with pro+mesopterygium and one with the metapterygium; pectoral and pelvic fins abutting; transversely elongated puboischiadic bar, similar to the condition seen in *Squatina* and *Rhinobatos*, but not as thin and thus more as in *Orectolobus*; puboischiadic bar without processes; anterior compound radial (i.e., anterior pelvic basal) directed laterally or slightly anterolaterally; occipital hemicentrum present; calcified vertebral centra; plate-like supraneurals; two dorsal fins

with basal plates supporting one fin spine each; ventral margin of basal plates in contact with the vertebral column; dorsal fin spines largely internal, lacking ornamentation and round in cross-section; precaudal hemal spines not as elongate as caudal basiventrals; anal fin absent; caudal fin leaf-shaped; and enlarged dermal denticles in front of the orbits and along a dorsomedial row on the trunk.

### 3.1.3. Description

**General features**: Specimen PBP-SOL-8007 represents a nearly complete, articulated female individual on a single slab without counterpart (Figures 1A and 3). The slab measures 125 cm in length, 60 cm in width and 3.5 cm in thickness. The specimen itself has a total length of 103 cm and is dorsoventrally flattened. It is exposed in dorsal view, but its unpaired fins are twisted to the side and preserved in left lateral view. Soft tissue (skin and muscle imprints) is preserved and provides information about the body outline. The cranium (neurocranium and splanchnocranium) is well-separated from the pectoral fins. The snout is triangular, and the head is longer (22 cm; 21.6% TL) than wide (17 cm; 16.5% TL). Pectoral fins are large (PFS = 46.1 cm; 44.7% TL) and rounded and insert close posterior to the head (PP1 = 25.9% TL). Both dorsal fins D1 and D2 originate well posterior to the pectoral girdle (PD1 = 47.4% TL, PD2 = 70.9% TL) and D2 is situated close to the caudal fin (DCS = 6.7% TL). All measurements taken follow Compagno [72] and are summarized in Table 1 (see also Figure 2). The anterior and posterior fin spines are well-preserved and are also exposed in left lateral view. Fossilized myotomal muscles are preserved posterior of the pectoral girdle on both sides of the vertebral column. A median row of enlarged placoid scales is present dorsally along the trunk. Between the median row of placoid scales and the pectoral fin, a lateral line organ moves along the body axis on either side. The lateral line on the left side extends from near the head region to the level of the second dorsal fin. On the right side of the body, the lateral line is less well-preserved and only visible at the level between the pectoral girdle and the pelvic fin. The lateral line is supported by C-shaped denticles in PBP-SOL-8007 (Figure 3), a feature that is also observable in the holotype.

**Neurocranium**: The neurocranium of PBP-SOL-8007 is completely preserved and exposed in dorsal view (Figures 3 and 4). The neurocranium extends forward over the subterminal mouth and overlies most of the mandibular arch (i.e., Meckel's cartilages and palatoquadrates). The anteriormost part of the neurocranium, the rostrum, forms a dorsally open, lateroventrally walled scoop and encompasses the precerebral fossa and the precerebral fontanelle anteriorly. A subnasal fenestra (basal communicating canal), a perforation penetrating the precerebral fossa ventrally, could not be identified in PBP-SOL-8007. Other specimens (e.g., MB 14-12-22-1), in which the ventral floor of the precerebral fossa is preserved, a subnasal fenestra is absent. The anteriormost part of the rostrum is not preserved in PBP-SOL-8007 and most other examined specimens, indicating that this portion of the rostrum was less mineralized than the remaining neurocranium. Rostral appendices are absent. Posterior to the rostrum are the paired nasal capsules, which are laterally expanded and mark the broadest part of the neurocranium. A small groove with a foramen is present on the posterior end of the nasal capsule on each side in PBP-SOL-8007. The position of the foramina in front of the orbital region indicates that they are the openings for the preorbital canal. No other foramina could be observed on the neurocranium of this specimen or in any of the other reviewed specimens. Posterior to the nasal capsules lie the orbits (lateral eye cavities). The orbits are roofed by a supraorbital crest. A suborbital shelf building the ventral floor of the orbits is missing. Due to the dorsal exposition of PBP-SOL-8007, the presence of a basal angle or the basitrabecular process could not be identified. The postorbital processes between the orbital and the otic region of the neurocranium are well-pronounced. On the posterior part of the neurocranium (otic region), otic bullae and, posterior to them, a depression interpreted as the endolymphatic fossa are visible. No foramen of the

endolymphatic and perilymphatic ducts could be identified. At the posteriormost part of the otic region, the foramen magnum opens and is placed at the center of the occiput. An occipital hemicentrum (equivalent to the posterior half of the calcified double cone that forms a complete vertebral centrum) is present and articulates laterally with the occipital condyle of the occiput.

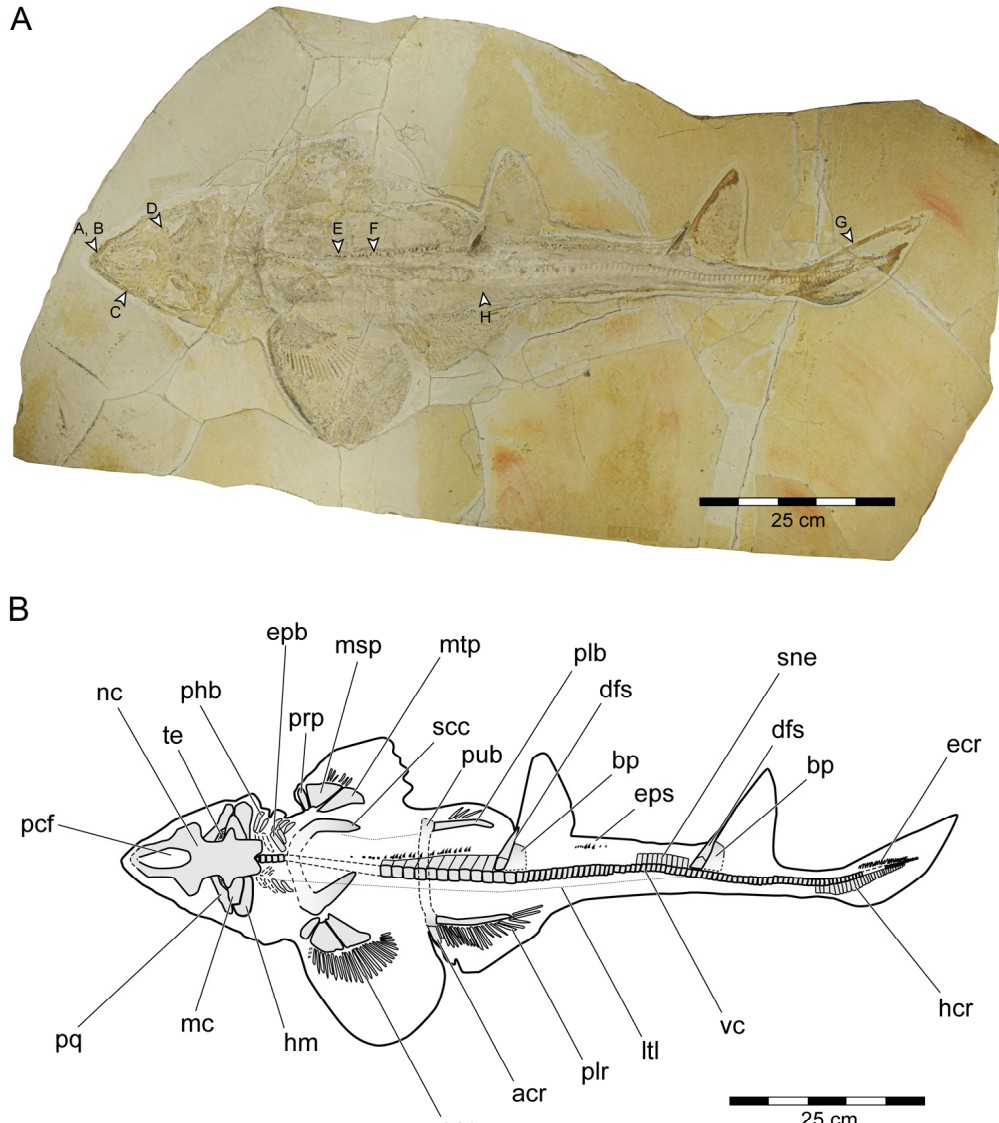

**Figure 3.** *Protospinax annectans* Woodward, 1918, PBP-SOL-8007, from the lower Tithonian of the Solnhofen Archipelago, Bavaria, Germany. (**A**) Slab containing the specimen. Arrowheads indicate the position of the dermal scales illustrated in Figure 8; (**B**) interpretative line drawing. *Abbreviations*: **acr**, anterior compound radial; **bp**, basal plate; **dfs**, dorsal fin spine; **ecr**, epichordal radials; **epb**, epibranchial; **eps**, enlarged placoid scales; **hcr**, hypochordal radials; **hm**, hyomandibula; **ltl**, lateral line; **mc**, Meckel's cartilage; **msp**, mesopterygium; **mt**, metapterygium; **nc**, neurocranium; **pcf**, precerebral fossa; **pcr**, pectoral radial; **phb**, pharyngobranchial; **plb**, pelvic basipterygium; **plr**, pelvic radial; **pq**, palatoquadrate; **prp**, propterygium; **pub**, puboischiadic bar; **scc**, scapulocoracoid; **sne**, supraneuralia; **te**, teeth; **vc**, vertebral column.

**Mandibular arch**: The mandibular arch includes the paired lower jaw cartilages (i.e., Meckel's cartilage) and the paired upper jaw cartilages (i.e., palatoquadrates). Both the Meckel's cartilages and the palatoquadrates of PBP-SOL-8007 are poorly preserved and mostly covered by the orbital region of the neurocranium (Figures 3 and 4). The mandibular arches are best preserved (but displaced) in JME-SOS 3386 (Figure S4) and in MCZ VPF-6394 (see de Carvalho and Maisey [47] figs. 7 and 9A). In JME-SOS 3386, the Meckel's cartilages and the palatoquadrates are disarticulated and separated from each other. The left palatoquadrate of the upper jaw is preserved as an imprint, while the right palatoquadrate and both Meckel's cartilages are complete. The right ramus of the palatoquadrate is preserved in labial view and lies close to the orbital region, with its posterior end partly covered by the neurocranium. Both rami of the palatoquadrate are united symphyseally. The left Meckel's cartilage is preserved in labial view, whereas the right ramus is preserved in lingual view. The anteriormost portions of both Meckel's cartilages are partly covered by the postorbital process of the neurocranium. The Meckel's cartilage is gradually tapering towards the symphysis. The palatoquadrate is slightly thinner than the Meckel's cartilage and bears a prominent orbital process mesially at its dorsal margin. The jaw suspension is thus of the orbitostylic type (sensu Maisey [182]).

No labial cartilages are preserved in PBP-SOL-8007. Labial cartilages are superficial to the mandibular arch and support the mouth corners. In SNSB-BSPG 1963-I-19, one dorsal and ventral labial cartilage is present on the left mandibular arch.

**Dentition**: Teeth of the Meckel's cartilage are preserved in their original, life position in PBP-SOL-8007, but are mostly covered by the neurocranium. Only a small patch of anterior teeth and more lateral tooth series are exposed (Figure 4D). No teeth of the palatoquadrate are preserved. The teeth of the lower jaw are small (up to 1 mm) and wider than they are high, with a dorsoventrally flattened and thin crown. A crest can be faintly expressed on the tooth crown, demarcating the lingual and labial faces. Both the lingual and the labial face are smooth and lack any ornamentation. The labial edge of the crown (i.e., visor) is convex and overhangs the root. The lingual face bears a well-developed uvula. The root is rather low with a flat base. A furrow is present on the basal face of the crown, separating the root into two lobes. A large central foramen opens in this furrow. Additional foramina are present on the labial face of the root and one to two foramina on each side of the uvula on the lingual face of the root.

The teeth in the other specimens are usually found disarticulated and dislocated, often far away from the respective jaw cartilage. In NHMUK PV P 37014, parts of the dentition are preserved intact but dislocated from the jaw cartilage, presumably because the teeth were firmly attached to the dental lamina and the underlying ectomesenchyme that became detached from the jaws (see Maisey [54] pl. 112). Specimen JME-SOS 3386 allows to deduct the original position of its teeth and suggests that *Protospinax annectans* had a rather homodont dentition and did not exhibit dignathic heterodonty.

An isolated tooth (EMRG-Chond-T-80) was micro-CT scanned to examine the tooth histology in *Protospinax annectans* (Figure 5). A large, hollow pulp cavity is present, mainly in the root but extending a short distance apically into the crown. The pulp cavity is connected to the outside via the foramina in the root. Circumpulpar dentine radiates from the pulp cavity into the crown and root. Osteodentine is absent, even in the root, a histotype that is rather uncommon in sharks and was previously described as "regular orthodont" in Pristiophoriformes and several batoid taxa [108]. The enameloid is barely discernible from the underlying dentine in the CT scans, which is most likely is due to re-mineralization during the fossilization process as examined in previous studies [107,108]. According to Thies [58], *Protospinax annectans* exhibits a three-layered enameloid.

**Hyoid arch and branchial arches**: The hyoid apparatus, consisting of a basihyal, paired ceratohyals (ventral), and paired hyomandibulae (dorsal), forms the second arch of the visceral skeleton and serves in jaw suspension and pharyngeal movement (i.e., opening and closing of the mouth). In PBP-SOL-8007, only the hyomandibula is preserved on either side, whereas the ceratohyal is not visible, possibly covert under the hyomandibula and the otic region of the neurocranium (Figures 3 and 4). The basihyal, an unpaired element articulating with the ceratohyal ventrally, is not preserved. The hyomandibula has a prominent distal end, which is articulating with the mandibular arch at the jaw joint between Meckel's cartilage and palatoquadrate, and tapers towards its proximal end which articulates with the otic region of the neurocranium.

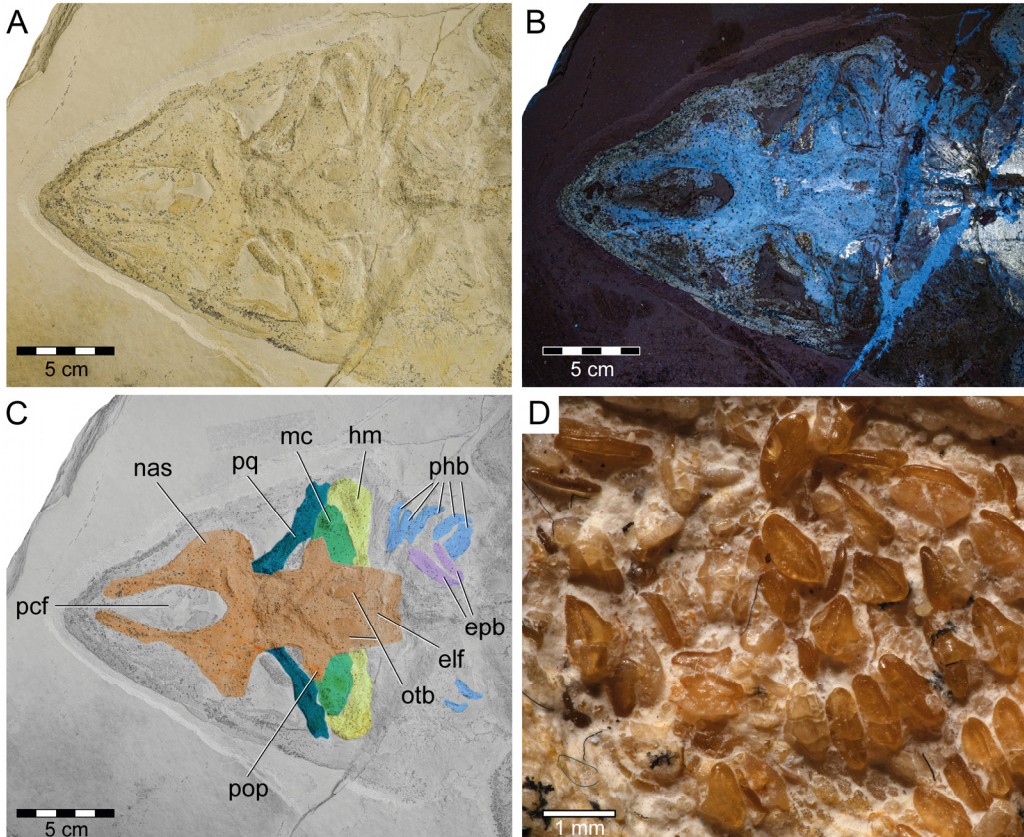

**Figure 4.** Head and teeth of *Protospinax annectans* Woodward, 1918, PBP-SOL-8007, from the lower Tithonian of the Solnhofen Archipelago, Bavaria, Germany. (**A**) Head region under normal light; (**B**) head region under UV light; (**C**) color-coded skull, showing the preserved skeletal elements of the neuro- and splanchnocranium; (**D**) close-up image of the teeth. *Abbreviations*: **elf**, endolymphatic fossa; **epb**, epibranchial; **hm**, hyomandibula; **mc**, Meckel's cartilage; **nas**, nasal capsule; **otb**, otic bullae; **pcf**, precerebral fossa; **phb**, pharyngobranchial; **pop**, postorbital process; **pq**, palatoquadrate.

Posterior to the hyoid arch lies the branchial apparatus with typically five to seven branchial arches in elasmobranchs. The branchial arches are composed of four connected elements, two dorsal elements (pharyngobranchial and epibranchial), and two ventral elements (ceratobranchial and hypobranchial). The hypobranchials are connected to the unpaired basibranchials and the basibranchial copula ventromedially. The branchial apparatus in PBP-SOL-8007 is incompletely preserved. Five branchial arches are preserved on the right side, whereas on the left side, only four arches are faintly discernible. Only the dorsal elements, the pharyngobranchials and the epibranchials are preserved, but the whole ventral part of the branchial apparatus, including the basibranchial elements, is missing. SNSB-BSPG 1963-I-19 is exposed in ventral view and the basihyal is preserved. The basihyal is chevron-shaped and slightly wider than it is deep.

**Pectoral girdle and fins**: The pectoral girdle (i.e., scapulocoracoid) in PBP-SOL-8007 is incompletely preserved and exposed in dorsal view (Figure 3). It is missing the ventral portion of the coracoid bar and it thus cannot be determined if the paired scapulocoracoids in PBP-SOL-8007 are separated and have a medial sternal cartilage lying in between (as in hexanchiform sharks) or if they are fused forming a single coracoid bar (as in most sharks and rays). Specimen SNSB-BSPG 1963-I-19, to our knowledge the only specimen of *Protospinax annectans* preserved in ventral view, clearly shows a single coracoid bar that is fused with the scapulocoracoid processes. The scapulocoracoid processes are flipped posteriorly so that their anterior side is exposed. The scapulocoracoid is U-shaped with the distalmost part of the scapular processes pointing posteromediodorsally in their natural position. The scapular processes are lacking dorsal suprascapulae and are not fused with the vertebral column. On the lateral face of the scapulocoracoid, slightly posterior, at the level of the junction of the coracoid bar and the scapular processes, articular surfaces (two condyles) for the pectoral basals are present. A discrete anterolateral condyle (promesocondyle) articulates with the pro- and mesopterygium of the pectoral fin, while the metapterygium articulates with a separate condyle (metacondyle). The articular surface on the scapulocoracoid is best seen in SNSB-BSPG 1963-I-19. No foramina or fenestrae are detectable on the scapulocoracoid.

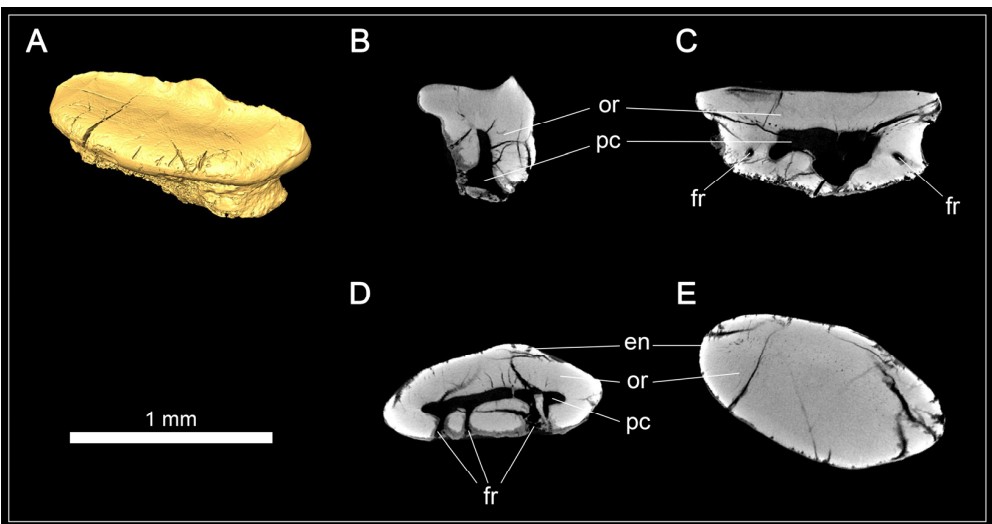

**Figure 5.** Micro-CT images and tooth histology of *Protospinax annectans* Woodward, 1918, EMRG-Chond-T-80, from the lower Tithonian of the Solnhofen Archipelago, Bavaria, Germany. (**A**) Isosurface; (**B**) virtual section of the tooth through the sagittal plane; (**C**) virtual section of the tooth through the frontal plane; (**D**) virtual section of the root through the axial plane; (**E**) virtual section of the crown through the axial plane. *Abbreviations*: **en**, enameloid; **fr**, foramen; **or**, orthodentine; **pc**, pulp cavity.

Both pectoral fins are preserved although to a different extent. The right pectoral fin is less well-preserved and only exhibits the three basal elements (propterygium, mesopterygium, and metapterygium) and six proximal radials (Figure 3). The three basal elements are well-separated and do not fuse with each other (Figures 3 and 6A,B). Meso- and metapterygium have a broad and rounded distal margin and are tapered towards the proximally situated articulation area, providing them an almost triangular shape. The propterygium, conversely, does not taper in any direction and is more or less rectangular in shape, and longer than wide. The propterygium is directed anteriorly but is not very extended and does not reach the level of the otic region of the neurocranium. The mesopterygium is the largest of the basal elements. In contrast to the left pectoral fin, the right pectoral one is very well-preserved, exhibiting three basal elements, 25 radials, and the outline of the fin shape. The right pectoral fin in PBP-SOL-8007 is large (pectoral fin length P1L = 18.5 cm/17.9% TL, pectoral fin span PFS = 46.1 cm/44.7% TL), aplesodic, and has a

rounded shape. It is unclear if the propterygium supports any pectoral radials, whereas the meso- and metapterygium are supporting nine and 15 radials, respectively.

**Pelvic girdle and fins**: The pelvic region of PBP-SOL-8007 is poorly preserved and the puboischiadic bar is mostly missing (Figure 3). Remains of the puboischiadic bar are detectable close to the left pelvic fin where it articulates with the basipterygium and the anterior compound radial (Figures 3 and 6C,D). The original position of the remaining puboischiadic bar can be recognized by an indentation in the matrix between both pelvic fins and inserts at vertebrae 25–26. The puboischiadic bar is mostly missing in PBP-SOL-8007, but is preserved to a variable degree in the other specimens, with MCZ VPF-6394 exhibiting the most complete pectoral girdle (see de Carvalho and Maisey [47] figs. 6 and 9C). The puboischiadic bar is transversely elongated and lacks any processes. Both the anterior and the posterior margins of the puboischiadic bar are straight, with only the medial part of the posterior margin being slightly arched anteriorly.

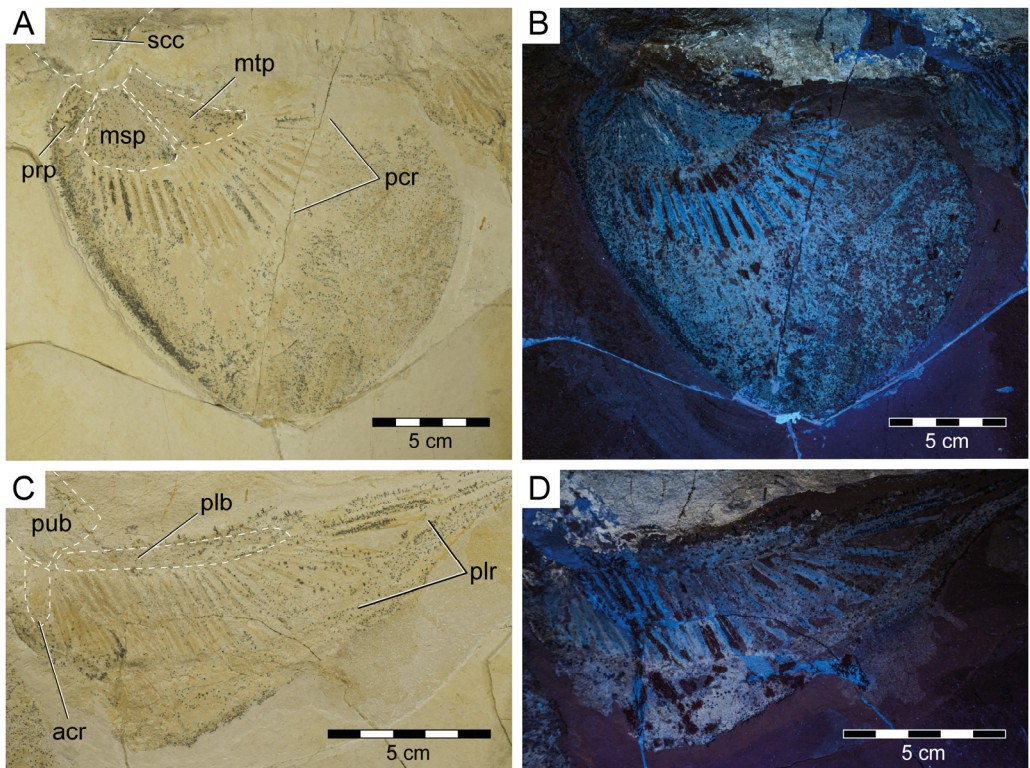

**Figure 6.** Pectoral and pelvic fins of *Protospinax annectans* Woodward, 1918, PBP-SOL-8007, from the lower Tithonian of the Solnhofen Archipelago, Bavaria, Germany. (**A**) pectoral fin under normal light; (**B**) pectoral fin under UV light; (**C**) pelvic fin under normal light; (**D**) pelvic fin under UV light. *Abbreviations*: **acr**, anterior compound radial; **msp**, mesopterygium; **mtp**, metapterygium; **pcr**, pectoral radial; **plb**, pelvic basipterygium; **plr**, pelvic radial; **prp**, propterygium; **pub**, puboischiadic bar; **scc**, scapulocoracoid.

The right pelvic fin of PBP-SOL-8007 is not very well-preserved and largely missing, whereas the left pelvic fin is in a well-preserved state. The basipterygium is almost straight, only faintly curved, and is supporting 25 pelvic radials (i.e., radial pterygiophores; Figure 6C,D). The anterior compound radial (i.e., anterior pelvic basal) is directed laterally to slightly anterolaterally. The pelvic fin skeleton is only supporting the fin base (aplesodic fin support). Specimen PBP-SOL-8007 is lacking claspers (i.e., mixopterygia), and thus PBP-SOL-8007 can be identified as a female individual. All reviewed specimens lacked claspers, except for MB 14-12-22-1 (Figure S5). In this specimen, only the right pelvic fin and clasper are preserved. The clasper is exposed in dorsal view. It is posteriorly attached to the basipterygium and is moderately long and slender, being about as long

as 17 adjacent centra. Claspers consist of several cartilage elements which can be divided into three groups: basal elements (intermediate element and beta (β-) cartilage), axial elements (ventral marginal, dorsal marginal, and axial cartilages; these elements are fused together), and terminal elements (dorsal terminal, ventral terminal, and accessory terminal 3 cartilages). The basal portion of the clasper in MB 14-12-22-1 is poorly preserved and it was not possible to identify any basal elements. Of the axial group, all elements are preserved. The axial cartilage is cylindrical and makes up most of the clasper, whereas the marginal and terminal cartilages are small and reduced. The dorsal marginal cartilage is almost triangular, with an elongate and very slender proximal part and a distally rounded part. The ventral marginal cartilage is to a large extent obscured by the dorsal marginal cartilage and the axial cartilage. Of the terminal group, two dorsal terminal cartilages and the accessory terminal cartilage three are exposed. The dorsal terminal cartilages articulate proximally with the dorsal marginal cartilage and are slender and slightly convex. The accessory terminal three cartilage articulates proximally with the ventral marginal cartilage and is thin and thorn-like in shape (Figure S5B,C).

**Axial skeleton and caudal fin**: The vertebral column of PBP-SOL-8007 is incompletely preserved and is missing a 12 cm long portion at the level of the pectoral girdle and a 3 cm long portion in the caudal fin region (Figure 3). One hundred and eight calcified vertebral centra are preserved. When the missing portions are taken into account, a total number of around 131 vertebrae is estimated (the vertebral number around the pectoral girdle was estimated by dividing the distance of the missing portion of the vertebral column by the average cranial–caudal length (ccl) of the three preceding and the three subsequent vertebrae (n = 15); the vertebral number of the caudal region was reconstructed based on the number of the hypochordal rays (n = 8)). Henceforth, when referring to locations, the reconstructed vertebral number is used as a reference point (e.g., first dorsal fin spine placed at the 36th vertebrae).

The calcification pattern of the vertebral centra was originally described as "probably tectospondylic" [53]. Maisey [54] disagreed and described the centra to be asterospondylic but did not provide further information about his assessment. No thin sections were taken from PBP-SOL-8007 and we thus cannot provide any information about the calcification pattern of the vertebrae. Parallel radials on the lateral surface of the vertebral centra, such as in Orectolobiformes and Lamniformes, are absent in *Protospinax annectans*.

The cranial–caudal length (ccl) of the first two vertebral centra is rather short (5.8 and 6.0 mm, respectively) and increases in the successive vertebrae between centra three to 32 (7.3 to 10.4 mm). From centrum 33 onwards, the ccl of the subsequent centra is declining (5.8 to 7.5 mm), followed by another drop between centra 92 and 105 (4.9 to 5.8 mm). The remaining centra gradually decrease in size (minimum ccl = 1.8 mm; see measurements in Table S1). Myosepta and imprints of the myotomal muscles in between the septa are well-preserved in most parts of the fossil, especially in front of the pectoral girdle and in the caudal region. The transition from monospondylous (one centrum per myomere) to diplospondylous vertebrae (two centra per myomere) occurs at the level of the first dorsal fin (between vertebrae 34 and 45), coinciding with the declining ccl of the vertebral centra. Due to the insufficient preservation of the myosepta in this area, it is not possible to further pinpoint this transitional zone, nor is it possible to determine how many vertebrae it includes. Vertebral ribs are not preserved in PBP-SOL-8007 but were identified in other specimens (Figure S4) and are well-documented in the literature (see de Carvalho and Maisey [47] Figure 3). The vertebral arches are not preserved, except for the posteriormost part of the caudal peduncle (between vertebrae 97 and 101) and the (dorsal) epaxial (epichordal rays) and (ventral) hypaxial skeletal elements (hypochordal rays) of the caudal fin (Figure 7). Of the epaxial skeletal elements, only the supraneural cartilages are detectable, whereas the neural arches (basidorsals and interdorsals) are not preserved. The supraneurals are expanded and plate-like in front of the first and second dorsal fin and in the abdominal region (Figure 3 and Figure S4). The hypaxial skeletal elements comprise basiventral cartilages (forming the hemal arches) and hemal spines, which are fused and

continuous in PBP-SOL-8007. None of the hemal spines in the caudal fin are detached from the basiventral cartilages, i.e., the caudal fin skeleton is undivided (no distinction between anterior and posterior diplospondylic region). The hemal spines (i.e., hypochordal rays) are flattened, stout, and display a rectangular outline, whereas the supraneural spines (i.e., epichordal rays) are slender, elongated, and terminally tapered (Figure 7).

The caudal fin is rather short and comprises 30 vertebral centra. It is leaf-shaped, lacking a distinct lower lobe and exhibits a very convex ventral margin, a weakly convex dorsal margin, and a pointed apex. Maisey [54] described a deeply notched lower lobe in the holotype specimen NHMUK PV P 8775, which previously was interpreted as an anal fin [53] (see discussion below). In contrast to the holotype, which exhibits a poor soft tissue preservation in the abdominal region, most of the specimens examined here have an exquisite soft tissue preservation and clearly demonstrate the lack of a distinct lower lobe.

**Unpaired fins**: Specimen PBP-SOL-8007 displays two moderately sized and angular-shaped dorsal fins (Figures 3 and 7). Both dorsal fins originate posterior to the pelvic girdle and close to the caudal fin, inserting at the 36th and between the 72nd and 73rd vertebrae, respectively. Each dorsal fin extends over 13 vertebrae. The first dorsal fin is slightly larger (8.4 cm/8.1% TL long; 12.5 cm/12.1% TL high) than the second dorsal fin (8.1 cm/7.8% TL long; 11.5 cm/11.1% TL high) (Table 1). The dorsal fins are almost completely covered by densely arranged dermal denticles that indicate their outline. Of the internal structures building up the dorsal fin, only the basal plate (i.e., basal pterygophore) is preserved in each dorsal fin; radials (or radial pterygophores) and ceratotrichia (i.e., derivates of the dermis) are not preserved in either of the dorsal fins. Each dorsal fin bears a single dorsal fin spine that is anchored anteriorly on the basal plate (Figure 7A–C).

An anal fin could not be detected in PBP-SOL-8007. Woodward [53] described a lobe near the caudal fin in the holotype as anal fin, which later was reinterpreted as a deeply notched hypochordal lobe of the caudal fin by Maisey [54] (see above). However, except for the holotype, none of the other specimens revised here (including PBP-SOL-8007) exhibits such a structure. In contrast to most of the other examined specimens, the abdominal region of the holotype specimen is poorly preserved, and the alleged "lobe" is here interpreted as detached soft tissue. We, therefore, regard this structure as a taphonomic artefact rather than an anal fin/hypochordal lobe. It should be noted though that another specimen, with a what appears to be a prominent anal fin, was illustrated in Frickhinger [175] (p. 90, fig. 163). However, the position of the supposed anal fin in this specimen is clearly detached from the caudal fin and thus differs from the state observed in the holotype specimen. The specimen illustrated in Frickhinger [175] is housed in an unknown private collection and was unavailable for studying, which is why we must regard the anal fin to be missing in *Protospinax annectans* until further proof of the opposite.

**Dorsal fin spines**: The two dorsal fin spines in PBP-SOL-8007 are exposed in left lateral view. They are stout, rounded in cross-section, and gently curved posteriorly (Figure 7A–C). The anterior fin spine is only partly preserved, missing the most basal part and the tip, and measures 44.5 mm in height. The posterior fin spine is completely preserved and measures 59 mm in height. The holotype and most other examined specimens of *Protospinax annectans* lack well-preserved dorsal fin spines. However, in the paratype (NHMUK PV P 37014), both fin spines are well-preserved and equal in size (1.5 cm). Both dorsal fin spines are deeply inserted (i.e., they are largely internal), with their basal portion (the trunk) terminating close to the vertebral column. A distinct junction separates the trunk (basal) from the mantle (distal). The surface of the mantle is smooth, devoid of any ornamentation, and is covered by a shiny layer of enameloid, whereas the trunk is devoid of an enameloid layer.

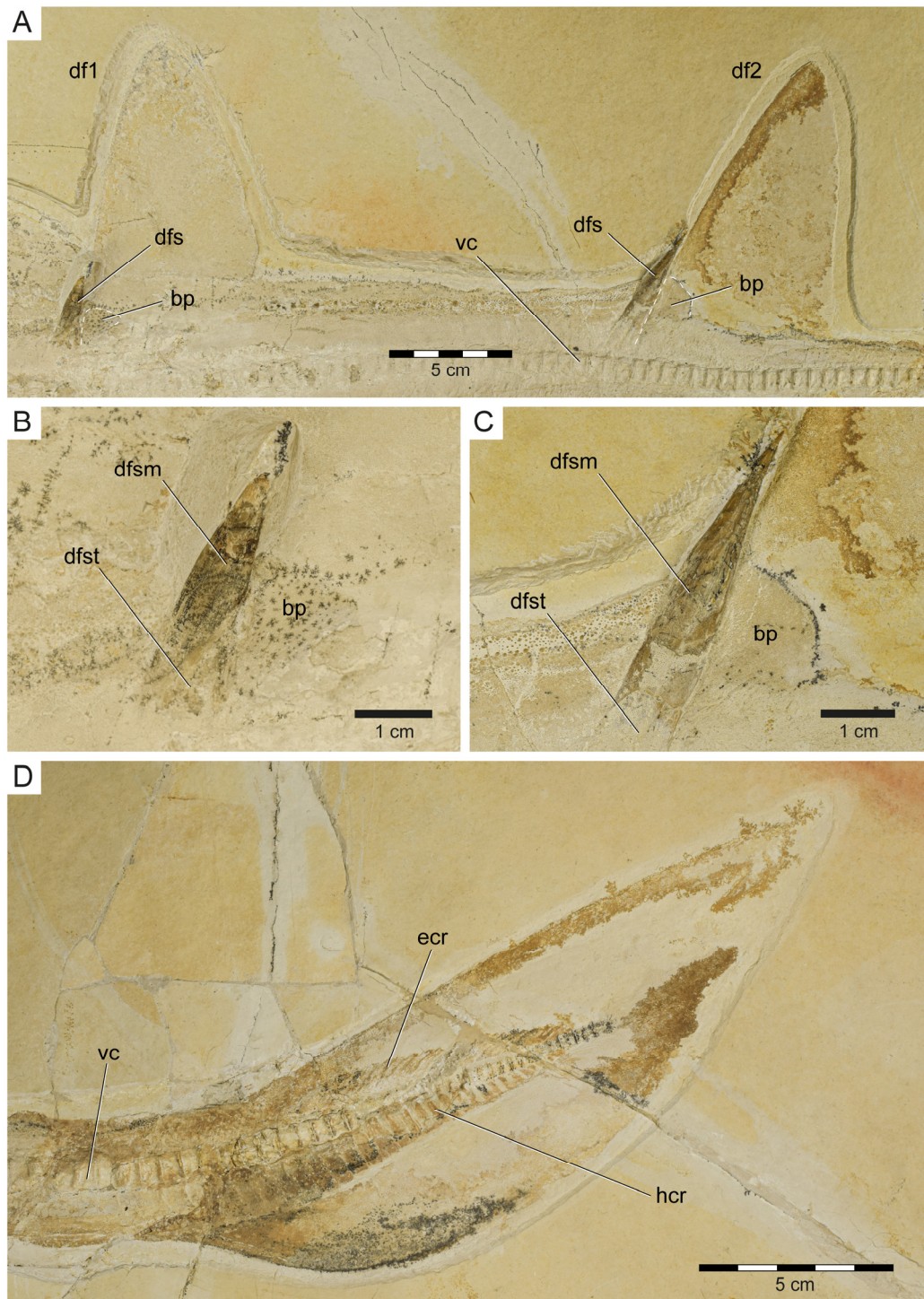

**Figure 7.** Dorsal and caudal fin of *Protospinax annectans* Woodward, 1918, PBP-SOL-8007, from the lower Tithonian of the Solnhofen Archipelago, Bavaria, Germany. (**A**) Dorsal fins with fin spines; (**B**) first dorsal fin spine; (**C**) second dorsal fin spine; (**D**) caudal fin. *Abbreviations*: **bp**, basal plate; **df1**, first dorsal fin; **df2**, second dorsal fin; **dfs**, dorsal fin spine; **dfsm**, dorsal fin spine mantle; **dfst**, dorsal fin spine trunk; **ecr**, epichordal radials; **hcr**, hypochordal radials; **vc**, vertebral column.

**Dermal denticles (Squamation)**: Most of the body of PBP-SOL-8007 is covered by dermal denticles (i.e., placoid scales), with patches of higher concentration of denticles being present in the head region (especially around the snout), anterior edges of the right pectoral fin and both dorsal fins, and on the dorsal margin of the caudal fin. In total, four different morphotypes could be detected in PBP-SOL-8007 (Figure 8): (1) denticles with an elliptical to round outline, bearing a large, asterisk-shaped central opening (Figure 8A,B); (2) denticles with a flat, smooth and knob-like crown (Figure 8C); (3) enlarged, thorn-like denticles with a circular base and a tapered crown (Figure 8D–F); (4) denticles with a prominent medial keel and two lesser pronounced lateral keels which are giving it an arrow-shaped outline in apical view (Figure 8G).

Morphotype 1 denticles are 250 µm in diameter and are restricted to the snout. Their morphology is undifferentiated, but they have a peculiar central opening which can be asterisk-shaped or more roundish in shape (Figure 8A,B). Similar dermal denticles have been described in the Late Cretaceous lamniform shark *Haimirichia amonensis* [183,184], which were thought to be associated with the electrosensory ampullary system. To the best of our knowledge, this morphotype has not been described in any other extant or extinct shark species.

Morphotype 2 denticles are smaller than type 1 denticles (up to 165 µm) and are restricted to the snout. Like type 1 scales, their morphology is rather undifferentiated, with a smooth knob-like crown (Figure 8C). They most likely serve a protective purpose (i.e., abrasion resistance) [185].

Morphotype 3 denticles are enlarged dermal denticles which are present in front of the orbits of the neurocranium and along a medial row on the dorsal surface of the trunk (Figure 8D–F), starting at the level of the pectoral girdle and gradually diminishing towards the second dorsal fin. The largest scales along the medial row are at the level between the pelvic girdle and the first dorsal fin. In the orbital region there is a single, large, and prominent enlarged dermal denticle (880 µm), whose crown is broken and only the circular base is preserved. It is surrounded by several smaller denticles (280–440 µm) of the same morphotype, of which not only the base, but also the crown is preserved. Similar dermal denticles have been reported for *Squatina* and several ray species [186,187].

Morphotype 4 is probably the most common morphotype in PBP-SOL-8007 and can be found almost on the entire body, with a high concentration at the anterior edge of the pectoral and dorsal fins, as well as on the dorsal edge of the caudal fin. The size can range quite significantly from 40 µm to 170 µm, with larger denticles being closer to the fin edge (Figure 8G). The differentiated morphology with three keels suggests that this morphotype serves hydrodynamic purposes (i.e., reducing drag by improving the flow of water over the body) [188,189].

*3.2. Phylogenetic Analysis*

Most of the nodes of the molecular tree used as molecular backbone constraint were well-supported by high posterior probability values (pp >70%). Only three nodes had a lower value and were thus not enforced on the subsequent analysis of our data matrix (conservative molecular backbone): (1) [*Carcharhinus* + *Sphyrna*], pp = 0.3554; (2) [*Echinorhinus* + *Squatina*], pp = 0.4387; (3) [[*Echinorhinus* + *Squatina*] + [*Pliotrema* + *Pristiophorus*]], pp = 0.6541 (Figure S1). The parsimony analysis with the applied molecular backbone resulted in 18 most parsimonious trees (MPTs) with a best score of 593 steps. Consistency index (CI) and retention index (RI) of the MPTs were 0.470 and 0.789, respectively.

In the strict consensus tree, the phylogenetic position of *Protospinax annectans* remained largely unresolved, and it fell into a polytomy with all other orders (Echinorhiniformes, Hexanchiformes, Squaliformes, Squatiniformes + Pristiophoriformes) within the superorder

Squalomorphii (Figure 9). In the majority-rule consensus tree, *Protospinax annectans* was recovered as a sister taxon to a clade consisting of Squatiniformes and Pristiophoriformes (Figure 10).

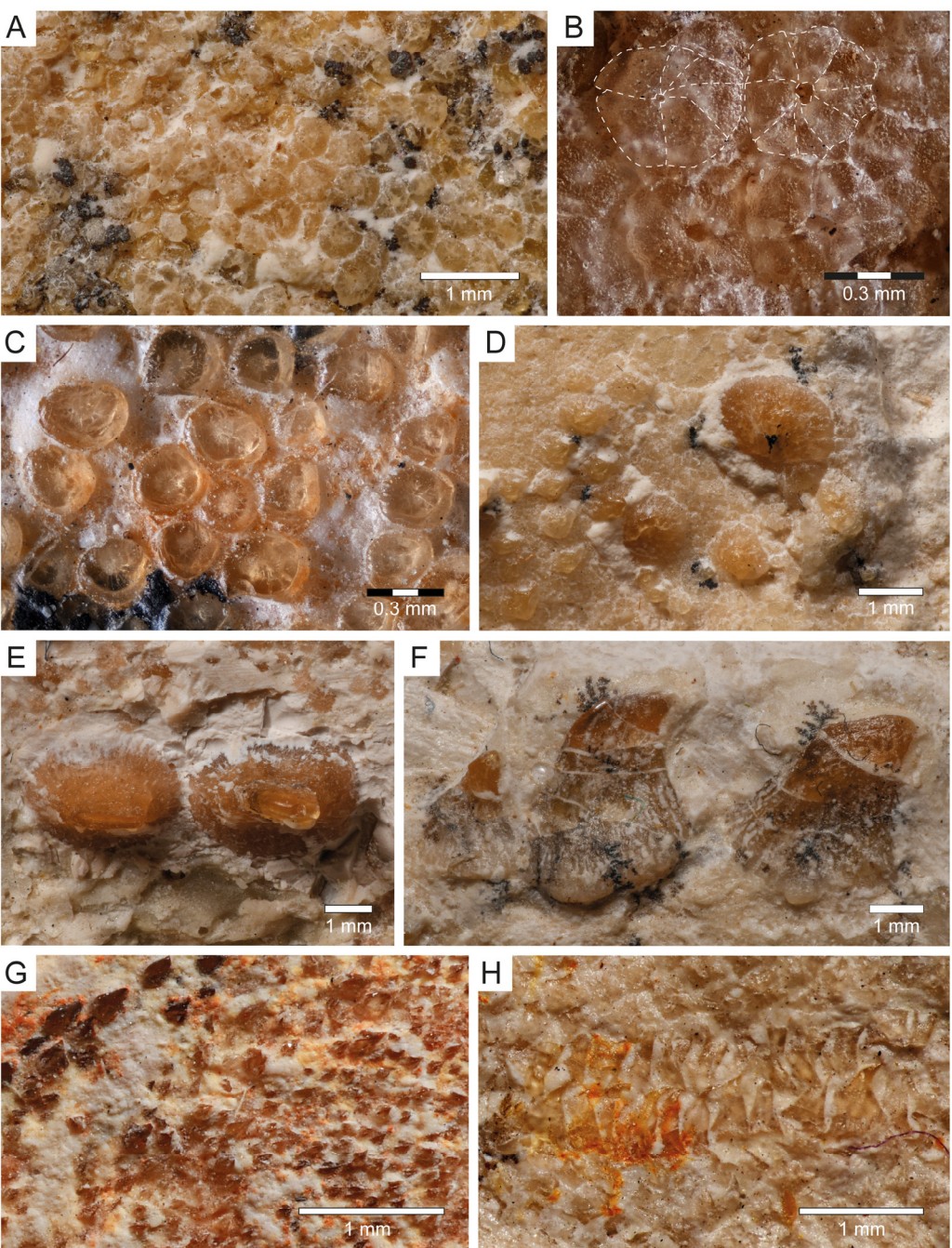

**Figure 8.** Dermal denticles of *Protospinax annectans* Woodward, 1918, PBP-SOL-8007, from the lower Tithonian of the Solnhofen Archipelago, Bavaria, Germany. (**A**) Morphotype 1; (**B**) close-up view of morphotype 1; (**C**) morphotype 2; (**D**) enlarged dermal denticles (morphotype 3) in front of the orbital region; (**E**) enlarged dermal denticles (morphotype 3) from the dorsal midline of the trunk in dorsal view; (**F**) enlarged dermal denticles (morphotype 3) from the dorsal midline of the trunk in lateral view; (**G**) morphotype 4; (**H**) C-shaped ringlets of the lateral line organ. The regions of the body, where the different morphotypes were extracted are summarized in Figure 3A.

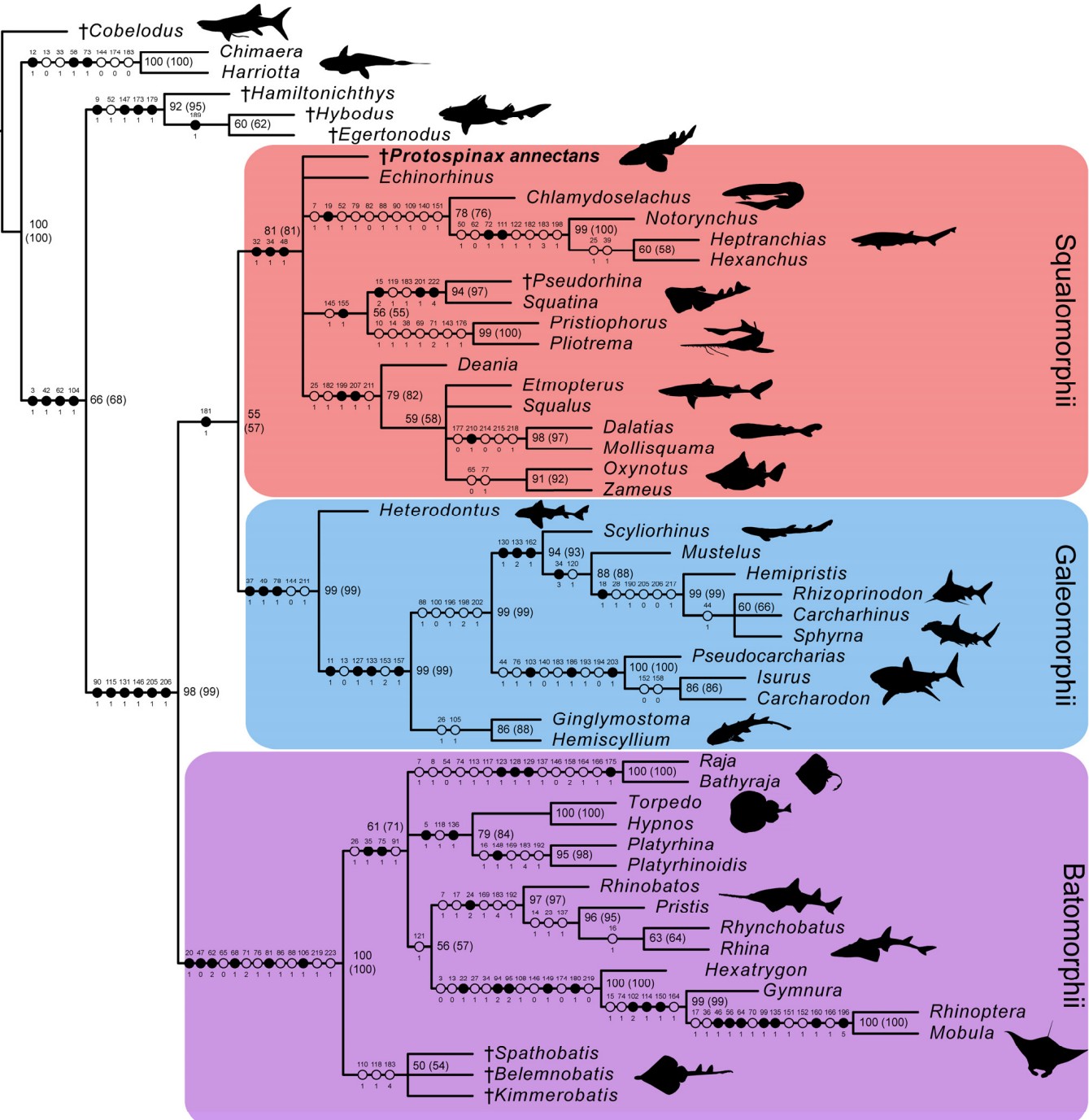

**Figure 9.** Strict consensus tree from the morphological data under a conservative molecular backbone constraint with mapped characters. Synapomorphies are indicated by black points, homoplasies are indicated by white points. Character number is provided above the point and character state is provided below. Numbers next to nodes represent the node support, bootstrap and jackknife values (jackknife values are in brackets). Daggers preceding taxon names denote extinct taxa. Silhouettes were either drawn by the authors (PLJ, MAS) or downloaded from www.phylopic.org (all downloaded images were available for reuse under the Public Domain Dedication 1.0 license).

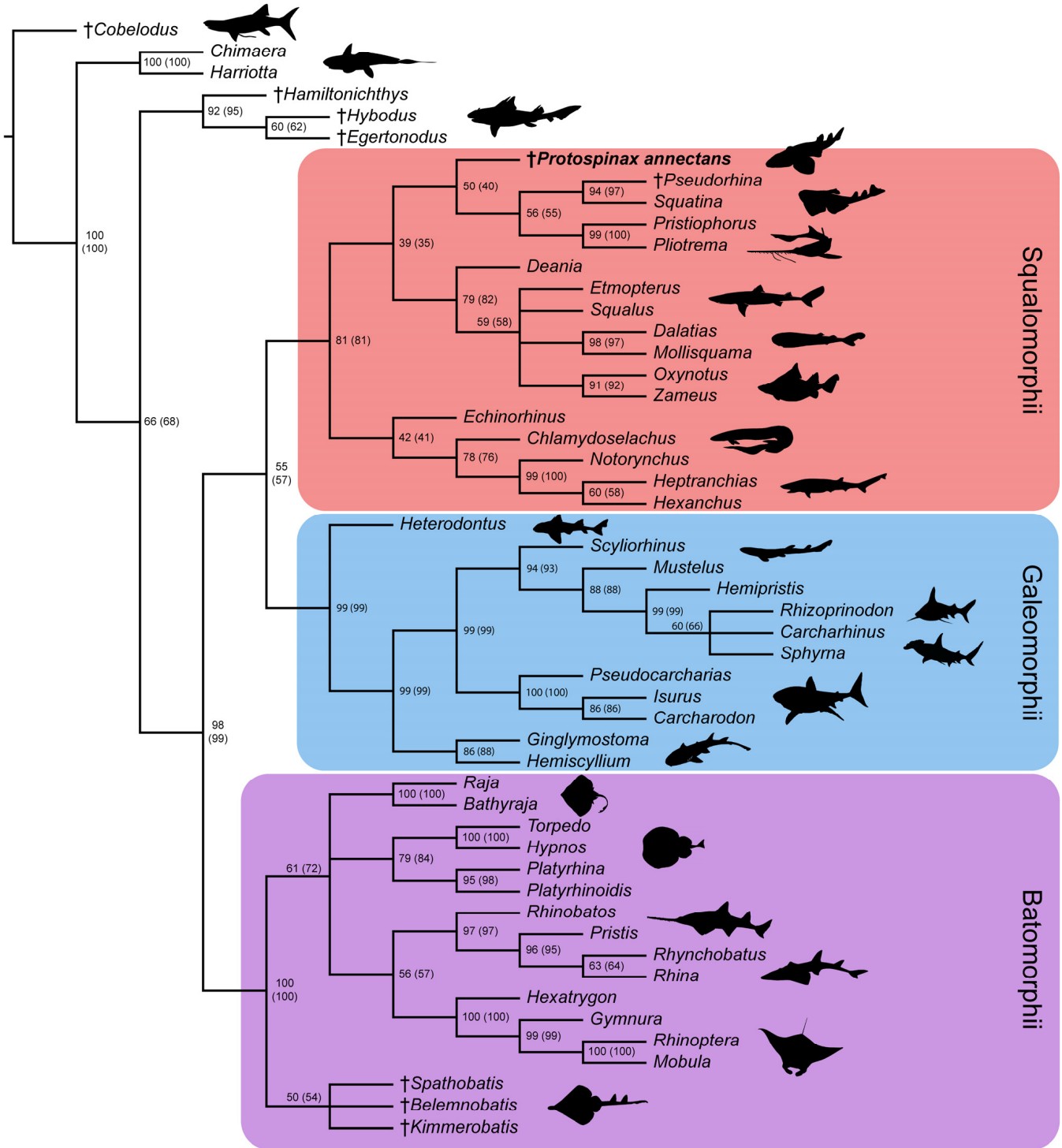

**Figure 10.** Majority-rule consensus tree from the morphological data under a conservative molecular backbone constraint with bootstrap- and jackknife frequencies (jackknife values are in brackets). Daggers preceding taxon names denote extinct taxa. Silhouettes were either drawn by the authors (PLJ, MAS) or downloaded from www.phylopic.org (all downloaded images were available for reuse under the Public Domain Dedication 1.0 license).

## 4. Discussion

### 4.1. Phylogenetic Implications for the Sister Group Relationship of Sharks and Rays

In our analysis with the conservative molecular backbone, the superorders Squalomorphii, Galeomorphii, and Batomorphii were recovered (note that even with a molecular

backbone constraint, only nodes that are supported by morphological data are recovered). As enforced by the molecular backbone constraint, the sister relationship between sharks (Squalomorphii and Galeomorphii) and rays (Batomorphii) was restored in our analysis, although with a very weak node support (Figures 9 and 10). The only synapomorphy that held the shark group together was the three-layered enameloid of their teeth (char. 181; Figure 9). The enameloid structure of modern shark teeth includes an external layer of single crystallite enameloid (SCE) and two interior components, the parallel bundled enameloid (PBE) and the tangled bundled enameloid (TBE), which together form the bundled crystallite enameloid (BCE) [190–192]. Heterodontiformes are the only modern shark order that deviates from this pattern: while their anterior teeth follow the pattern typical for modern sharks, their lateral teeth lack PBE [190,192]. Heterodontiform sharks exhibit a strong monognathic heterodonty with small labio-lingually flattened anterior teeth which can have one or more cusps (clutching-type teeth), and large, apico-basally flattened lateral teeth (grinding-type teeth) [52,108]. The absence of PBE in lateral teeth of this group is regarded as a secondary loss and an adaptation to durophagy [190,192]. In batomorphs and other chondrichthyan groups, it has long been proposed that their tooth enameloid only consists of the SCE unit (or lack an enameloid cap completely [193,194]), as opposed to the three-layered condition observed in modern sharks [192]. Although a single crystallite enameloid appears to be the most common condition in extant batomorphs [195], recent studies indicate that the enameloid microstructure in batomorphs might be more complex than previously thought [191,195,196]. Studies on one of the oldest batomorph groups, the Early Jurassic Toarcibatidae (formerly known as "Archaeobatidae"; see Greenfield et al. [197]), revealed the presence of a SCE and BCE unit in the teeth of *Toarcibatis*, *Cristabatis*, and *Doliobatis* (note that the latter genus was proposed to belong to the family Rhinobatidae instead [52]). Although the compaction of the bundles and the arrangement of the PBE and TBE components differ between this ancient batomorph family and modern sharks [198], it demonstrates that a multilayered enameloid was most likely present in the common ancestor of sharks and rays already. The presence of a multilayered enameloid in Synechodontiformes [199], the Permian Ctenacanthiformes *Neosaivodus flagstaffensis* Hodnett, Elliott, Olson and Wittke 2012 [200,201], and putative Ctenacanthiformes from the Lower Cretaceous of France [201] and Austria [202], further complicates the search for an unambiguous synapomorphy to define a shark group comprising Squalomorphii and Galeomorphii.

Despite the lack of unambiguous morphological data to support a sister group relationship between sharks and rays [45–48], several molecular analyses strongly favor this hypothesis [13,37,39,42,43,61,203]. However, alternative hypotheses also were retrieved by molecular studies (depending on the used genes and applied evolutionary models), for example with Galeomorphii being more closely related to Batomorphii, a shark paraphyly was also retrieved by molecular data (see Li et al. [204]). Although a sister-group relationship between modern sharks and rays was implied in the past [205–209], such a relationship only was retrieved once in phylogenies based on morphological data [59]. In the study of Villalobos-Segura et al. [59], monophyly of sharks was supported by two additional characters, the presence of labial cartilages (char. 72) and a short clasper length (char. 126).

Our analysis also included the presence and configuration of labial cartilages as characters (chars. 86–89). However, the presence of labial cartilages was not recovered as a synapomorphy for sharks in our analysis. Although labial cartilages were reported in many modern shark species [103,140,210–212], they are also well-known from Hybodontiformes [211,213–215], Holocephali [211] as well as Paleozoic chondrichthyans such as *Helicoprion* [216], *Trisychius* [217], and *Debeerius* [218]. The presence of labial cartilages in such a variety of different chondrichthyan taxa, paired with the fact that many groups are predating modern sharks indicates that the reconstruction of labial cartilages as a synapomorphy for modern sharks in Villalobos-Segura et al. [59] was merely a product

of outgroup choice and taxon sampling (which focused on batomorphs, in which labial cartilages are only known from Torpediniformes and Rhinopristiformes [59,87,219,220]).

The third proposed synapomorphy for modern sharks in Villalobos-Segura et al. [59], clasper length, was based on Aschliman et al. [87] (char. 73), who unfortunately did not provide a clear definition for this character. According to them, elongated and slender claspers are present in Chimaeridae, Rhinopristiformes, and Rajiformes, whereas the claspers of the remaining batomorphs and the outgroup (i.e, modern sharks) are relatively short and stout. According to the character optimization in Villalobos-Segura et al. [59], short claspers were recovered as a synapomorphy in sharks with independent gains in *Harriotta*, and a clade comprising Torpediniformes and Myliobatiformes. However, when comparing the clasper length of certain lamniform sharks (e.g., the white shark *Carcharodon carcharias*, the megamouth shark *Megachasma pelagios*, the basking shark *Cetorhinus maximus* [221]) with those of Rhinopristiformes, no clear difference is apparent in clasper length (even when considering the total length). We, thus, decided to exclude this character from our analyses, as a clear definition is needed (e.g., relative clasper length regarding TL) to determine if claspers can be regarded as "short" or "long". Without such a definition, it is difficult to test for the phylogenetic significance of this character.

### 4.2. Phylogenetic Implications for the Superorders Batomorphii, Squalomorphii, and Galeomorphii

Our analysis with the conservative molecular backbone recovered the three superorders Batomorphii, Squalomorphii, and Galeomorphii with relatively high node supports (Figures 9 and 10). Although the unconstrained tree resolved a paraphyletic shark group (i.e., Hypnosqualea hypothesis), it is worth mentioning here that high node support values for Galeomorphii and Batomorphii were also retrieved in this analysis, whereas the Squalomorphii and the incorporated "Hypnosqualea" clade only showed weak node support (Figure S2).

The superorder Batomorphii is very well-supported by a number of synapomorphies, five of which were unique to this group (Figure 9): the presence of an antorbital cartilage (char. 20); lack of a craniopalatine articulation (char. 47); hyoid arch with reduced ventral parts (ceratohyal absent) and a well-developed hyomandibula that is suspending the lower jaw directly (char. 62); last ceratobranchial articulates with the scapulocoracoid (char. 81); and an anteriorly extending propterygium (char. 106). Within Batomorphii, the Jurassic rays *Belemnobatis*, *Kimmerobatis*, and *Spathobatis* form a monophyletic group at the base of this group, as the sister group to all other batomorphs. A similar topology was retrieved in previous studies [59,95], although a closer relationship with modern Rhinopristiformes was suggested by others [90,94]. The remaining orders Myliobatiformes, Rajiformes, Rhinopristiformes, and Torpediniformes also are retrieved by our analysis, although their interrelationships remain unresolved apart from a sister relationship between Rhinopristiformes and Myliobatiformes, suggesting that the topology proposed in the molecular backbone is not supported by any morphological data. It is important to point out here that this is not solely a conflict between morphological and molecular data; higher group interrelationships among batomorphs remain controversial even when only molecular evidence [13,37,43,222] or morphological evidence [59,87,94–96] is considered.

The superorder Galeomorphii is supported by three unambiguous synapomorphies (Figure 9): hyomandibula fossa situated anteriorly in the otic region (char. 37); ethmoidal articulation (char. 49); and presence of a pharyngobranchial blade (char. 78). The main problem in this group was to determine the interrelationships between *Rhizoprionodon*, *Sphyrna*, and *Carcharhinus*. According to our molecular backbone tree, *Rhizoprionodon* is sister to a clade consisting of *Carcharhinus* and *Sphyrna*, although with a weak node support (Figure S1). Conflicting results can also be found in other molecular studies, in which either a *Carcharhinus-Sphyrna* clade [43] or a *Carcharhinus-Rhizoprionodon* clade [37] is proposed. More molecular and morphological studies seemingly are needed to resolve the topology within this clade and properly define the families Carcharhinidae and Sphyrnidae.

The superorder Squalomorphii is supported by three unambiguous synapomorphies (Figure 9): presence of a basitrabecular process (char. 32); a triangular-shaped postorbital process (char. 34); and orbitostylic articulation (char. 48). This superorder showed a significantly lower node support than the other two (Figures 9 and 10) and several of the nodes within this group showed weak node supports in the majority consensus tree (Figure 10). In the strict consensus tree, the interrelationships between the orders remained unresolved except for a suggested sister relationship between Squatiniformes and Pristiophoriformes (Figure 9). The most controversial taxon in our analyses is *Echinorhinus*. Morphologically, this taxon was usually associated with either Squaliformes [45,72] or Hexanchiformes [223,224]. The lack of an anal fin suggests a closer relationship with Squaliformes, whereas the dentition points towards a closer relationship with Hexanchiformes [52]. De Buen [225] suggested that the family Echinorhinidae should be considered as a separate order (Echinorhiniformes), which is supported to some extent by morphological [45] and molecular analyses [37,226]. In most molecular analyses, however, *Echinorhinus* is closely allied with Squatiniformes and Pristiophoriformes, although usually with weak node supports [13,37,41,61]. Contrary to earlier studies in which a sister group relationship between Echinorhiniformes and a clade comprising Pristiophoriformes and Squatiniformes was proposed, our molecular backbone suggests a basal position of Pristiophoriformes to a clade comprising Echinorhiniformes and Squatiniformes (Figure S1). However, the node support for these groups is rather low and they were not constrained but were allowed to fall within the phylogenetic tree based on the morphological data. In our analysis, *Echinorhinus* is positioned at the base of the Hexanchiformes, whereas Pristiophoriformes and Squatiniformes form a clade representing the sister group to the Squaliformes. The node support for these groups is low (Figure 10), although the pairing of Pristiophoriformes and Squatiniformes retrieved in our tree is well-supported by previous phylogenetic studies, both molecular [37,39,43,61,203] and morphological [45,46,48,78]. In the strict consensus tree, the phylogenetic interrelations of *Echinorhinus* with the other squalomorph orders remain unresolved, further supporting earlier claims that *Echinorhinus* should be placed within its own order. The position of *Echinorhinus* within Squalomorphii remains problematic at the moment as no morphological data supports the close relationship between Echinorhiniformes, Pristiophoriformes, and Squatiniformes. Further morphological and molecular studies are needed to settle the phylogenetic position of Echinorhiniformes and bring clarity to our understanding about the higher-level interrelationships within Squalomorphii.

### 4.3. Phylogenetic Interrelationships of Protospinax annectans with Comments on the "Hypnosqualea" Clade

The phylogenetic interrelationships of *Protospinax annectans* were examined cladistically based on skeletal characters in two previous studies, resulting in conflicting results. In the first study conducted by de Carvalho and Maisey [47], *Protospinax annectans* was restored as a basal hypnosqualean, a clade comprising Squatiniformes, Pristiophoriformes, and Batomorphii. The hypnosqualean hypothesis proposes a shark paraphyly, in which batomorphs are deeply nested within the squalomorph sharks [44–46,48].

The second phylogenetic analysis that incorporated *Protospinax annectans* was conducted by Villalobos-Segura et al. [59], which resolved a sister relationship between sharks and rays as proposed by molecular studies [37,39–43] (also see discussion in 4.1. Phylogenetic implications on the sister group relationship of sharks and rays). In that study, which focused on the interrelationships of fossil and extant Batomorphii and included several squalomorph and galeomorph sharks as outgroups, *Protospinax annectans* was recovered as an extinct sister group to Squaliformes [59].

In the following sections, we first discuss the hypnosqualean characters proposed by Shirai [44] and then discuss and compare the characters that led to the topology in de Carvalho and Maisey [47] and Villalobos-Segura et al. [59], and compare their results with our outcomes. This should help the reader to better understand our choice of characters

and character states in the present study before discussing the implications of our own results for the systematic position and phylogenetic relationships of *Protospinax annectans*.

### 4.3.1. Revision of the Hypnosqualean Characters Proposed by Shirai [44]

When de Carvalho and Maisey [47] examined *Protospinax annectans*, the hypnosqualean hypothesis was then the prevailing theory about elasmobranch interrelationships. In the original work of Shirai [44], the clade "Hypnosqualea" (Squatiniformes, Pristiophoriformes, Batomorphii) was supported by the following morphological characters: (1) absence of a basioccipital fovea; (2) occipital hemicentrum reduced (Squatiniformes) or absent (Pristiophoriformes and Batomorphii); (3) presence of a separate condyle for the propterygium; (4) enlarged plate-like supraneurals; (5) inclinator dorsalis arises directly from the vertebra; (6) dual ball and socket articulation between hyomandibula and auditory region; (7) flexor caudalis extending anteriorly into the precaudal region; (8) and coraco-hyomandibularis present [44]. Characters 5–8 could not be examined in *Protospinax annectans* and thus were left out in our analysis.

*(1) Basioccipital fovea:* The basioccipital fovea is a median concavity ventral to the foramen magnum, which in modern sharks is filled by the occipital hemicentrum. In previous phylogenetic studies, the presence of an occipital fovea was established to be the plesiomorphic condition in elasmobranchs and its absence in Squatiniformes, Pristiophoriformes, and Batomorphii was considered a synapomorphy for hypnosqualeans [44,45,47]. In *Protospinax annectans,* a basioccipital fovea is present, which supported its placement as a basal hypnosqualean [47]. However, this view is challenged by more recent morphological studies on the occipital region of Squatiniformes conducted by Claeson and Hilger [227] and de Carvalho et al. [228], which demonstrate the presence of an occipital fovea in fossil and extant Squatiniformes. Claeson and Hilger [227] further discuss that a prominent notch in the posterior basicranium ventral to the foramen magnum in Jurassic Batomorphii may be homologous with the basioccipital fovea in sharks (also see discussion in Maisey et al. [229] p. 22–23). Our reexamination of this character revealed that a prominent notch is also present in several extant batomorphs (e.g., *Raja*, *Torpedo*, and *Rhina*) and a basioccipital fovea is apparently also present in Pristiophoriformes. In the light of the uncertainty, whether the basioccipital fovea in modern sharks (including Pristiophoriformes) and the notch in the basicranium of basal and derived batomorphs are homologous structures or not, we decided to remove this character from our matrix and urge future studies to examine the development and homology of these structures in sharks and rays.

*(2) Occipital hemicentrum***:** The occipital hemicentrum is the morphological equivalent of the posterior half of the double-coned vertebral centrum, which is embedded in the basioccipital fovea and articulates with the first cervical centrum of the vertebral column [45,150]. The presence or absence of this feature was noted to be a distinctive difference between sharks (occipital hemicentrum present) and rays (occipital hemicentrum absent) [56,230,231]. The reduction in this structure in Squatiniformes and its loss in Pristiophoriformes was regarded as evidence for a subsequent loss of this structure in hypnosqualeans [44]. In subsequent phylogenetic analyses [46,48,77,78], the absence of an occipital hemicentrum also was coded for *Echinorhinus* and hexanchid sharks (*Notorynchus*, *Hexanchus,* and *Heptranchias*). However, Maisey [130] provided evidence for the presence of a poorly mineralized hemicentrum in *Notorynchus cepedianus* and Maisey et al. [231] again emphasized that an occipital hemicentrum was present in sharks and was lost only in rays. They further mention the presence of an occipital hemicentrum in embryonic pristiophorids (unpublished data [227]). Our reexamination of this character in CT-scans of *Pristiophorus* confirms this observation and thus, the absence of an occipital hemicentrum is regarded as being confined to the batomorphs. However, there is evidence that primitive batomorphs retained an occipital hemicentrum, as exhibited by a yet undescribed Early Jurassic ray from Germany mentioned in Maisey et al. [229] (this specimen is currently described in detail by Duffin and Kriwet).

*(3) Separate condyle for the propterygium*: Previously, it was thought that most elasmobranchs had a single condyle on the scapulocoracoid supporting the pectoral basals, with the exception of Squatiniformes having a separate condyle for the metapterygium and Pristiophoriformes and Batomorphii having a third condyle for the propterygium [44–46,48,77,78]. However, more recent morphological studies on the pectoral girdle of elasmobranchs revealed a much more complex structure of the scapulocoracoid in many groups [133]. Articulation with the pectoral fin can occur through a different number of condyles and facets on the scapulocoracoid. A separate procondyle was also found in *Etmopterus* and several lamniform sharks, although only two condyles are present in these taxa, with the second condyle articulating with the meso- and metapterygium. The articulation by three separate condyles on the scapulocoracoid was only reported for Pristiophoriformes and Batomorphii [133]. We included this new information into our analyses and did not find any unambiguous synapomorphies for any of the groups. In *Protospinax annectans*, two condyles articulate with the pectoral basals, one with the pro+mesopterygium and the metapterygium, respectively, such as in Squatiniformes.

*(4) Enlarged plate-like supraneurals*: Expanded supraneurals of the vertebral centra were reported anterior to the second dorsal fin in *Squalus* [45,232], as well as in Squatiniformes, *Pristiophorus*, and several batomorph species (Rhinopristiformes and Torpediniformes) [45,232], in which the expanded supraneurals extend to the abdominal and precaudal region. *Protospinax annectans* also exhibits expanded supraneurals along the abdominal region. It is tempting to regard this feature as an adaptation to the benthic lifestyle in elasmobranchs. However, it is unknown from benthic galeomorph sharks and thus is regarded here as an independent gain in Rhinopristiformes, Torpediniformes, and within Squalomorphii.

### 4.3.2. *Protospinax* as a Basal Hypnosqualean as Proposed by de Carvalho and Maisey [47]

In de Carvalho and Maisey [47], the following nine characters supported the phylogenetic position of *Protospinax annectans* within the "Hypnosqualea" (character numbers represent the numbering in Appendix 4 of de Carvalho and Maisey [47]): (1) subnasal fenestra absent (char. 6); (2) no basal angle (char. 22); (3) pelvic fins adjacent to pectorals (char. 33); (4) transversely elongate pelvic girdle (char. 35); (5) diagonal tooth arrangement (char. 36); (6) metapterygial condyle present (char. 61); (7) condition of anterior pelvic basal (char. 67); (8) plate-like supraneurals in abdominal region (char. 70); and (9) dorsal basal plates anchored to the vertebral column (char. 75).

*(1) Subnasal fenestra*: The subnasal fenestra is a ventral perforation at the proximal end of the precerebral fossa [45] and was originally thought to represent a squaliform synapomorphy by Compagno [233]. Shirai [45], however, noted that a subnasal fenestra is also present in the hexanchid sharks *Heptranchias* and *Hexanchus* (but not *Notorynchus*) and interpreted it as independent gains in both groups. The absence of the subnasal fenestra in Squatiniformes, Pristiophoriformes, Batomorphii, and *Protospinax annectans*, which were recovered as highly derived squalomorph sharks in de Carvalho and Maisey [47], was regarded as a secondary loss and a synapomorphy uniting this group. Our revision of this character in *Protospinax annectans* confirms the absence of a subnasal fenestra, and thus supports a closer relationship with Pristiophoriformes and Squatiniformes.

*(2) Basal angle*: It was reported in several shark groups and in Batomorphii that the trabecular-parachordal angle that is present in embryos straightens during development. In Squaliformes, hexanchids, and in the batomorph taxon *Pristis*, the angle between the trabeculae and parachordals becomes reversed and causes the basal angle [129,234,235]. The basal angle is a hump at the ventral floor of the braincase situated mesially to the orbital articulation [47]. Shirai [45] claimed that this feature was present in all Squalomorphii (also in *Chlamydoselachus*, Pristiophoriformes, and Squatiniformes) and in the batomorph taxon *Pristis*, as he regarded the "presphenoid bolster" described by Holmgren [232] to represent a basal angle. De Carvalho and Maisey [47] disagreed with this claim, referring to their own observations and regarded the basal angle to be absent in *Chlamydoselachus*, Squatiniformes, and Pristiophoriformes. We were not able to detect a basal angle in the examined CT-scans of these taxa

either, nor did scientific illustrations indicate the presence of such an angle [45,228,236,237]. It is worth noting though that Iselstöger [238] described a fold ("Basalecke") in the basicranium of *Squatina* within the floor of the precerebral fontanelle which was not regarded homologous to the basal angle seen in Hexanchidae and Squaliformes in subsequent studies [46–48]. A similar structure was also described in the galeomorph shark *Orectolobus japonicus* [232], indicating that the angle observed in both taxa may not refer to the basal angle seen in hexanchids and squaliform sharks. We acknowledge that further research is needed to resolve the nature of these structures observed in the neurocranium of *Squatina* and *Orectolobus* and the basal angle in squaliform and hexanchid sharks, but we tentatively follow the coding of previous studies [46–48,77,78] that regarded this structure to be absent in Squatiniformes and Pristiophoriformes. De Carvalho and Maisey [47] reported the absence of this structure in *Protospinax annectans*, although it is unclear how they came to this conclusion, as only one specimen (SNSB-BSPG 1963-I-19) is preserved in ventral view and they apparently only examined SNSB-BSPG 1963-I-19 based on the description and figures provided in Maisey [54], who did not mention the presence or absence of the basal angle in this specimen. Our reexamination of SNSB-BSPG 1963-I-19 revealed that the portion of the neurocranium that would bear the basal angle was mostly covered in matrix and obscured by the mandibular arch, not allowing to draw any unequivocal conclusions about the presence or absence of this structure in *Protospinax annectans*. We, therefore, left this character as "?" for *Protospinax* in our data matrix.

(3) *Pelvic fins adjacent to pectorals*: It was noted that *Protospinax annectans* and Squatiniformes had pectoral fins adjacent to the pelvic fins, whereas they were clearly separated from each other in Pristiophoriformes and supposedly primitive batomorphs (Rhinopristiformes). The presence of this feature in higher batomorphs was interpreted as a secondary gain [47]. This character is indeed rare in sharks and, to our knowledge, is restricted to Squatiniformes and *Protospinax annectans*, thus meriting its use as a taxonomic character. However, it is important to point out here that this condition is the result of the greatly enlarged pectoral fin in Squatiniformes, *Protospinax,* and several batomorphs and most likely corresponds to a specialized benthic lifestyle. The phylogenetic relevance of this character is ambiguous, which is why we excluded this character from our analyses.

(4) *Transversely elongate pelvic girdle*: The shape of the puboischiadic bar was described as being thin and wide in Squatiniformes, Rhinopristiformes (previously interpreted as "ancient rajiform condition") and *Protospinax annectans* [47]. Our reexamination confirms that Squatiniformes and Rhinopristiformes have a wide and thin pelvic girdle, but this can only be partly applied to *Protospinax annectans*, in which the puboischiadic bar is wider than in many other shark taxa, but it is not significantly thinner. We could observe a similar state in *Orectolobus maculatus*, which also has a rather wide but not very thin puboischiadic bar. Other taxa, such as *Ginglymostoma cirratum* and *Chiloscyllium punctatum* also are described as having a transversely elongate puboischiadic bar, although not as elongate as in "hypnosqualeans" [47]. It is apparent with all these uncertainties that a clear definition is lacking for this character and its phylogenetic significance is ambiguous. We, therefore, decided to exclude this character from our analysis, as the overall shape of the pectoral girdle, including the absence/presence of processes, differs greatly between *Protospinax annectans* and Squatiniformes/Rhinopristiformes and the phylogenetic relevance of the relative width of the puboischiadic bar is questionable.

(5) *Diagonal tooth arrangement:* The diagonal tooth arrangement (or alternate tooth arrangement [239–241]) is present in all elasmobranchs, except for Echinorhiniformes, Hexanchiformes, and Squaliformes, in which the teeth are arranged linearly [45,47,61,97].

Due to the phylogenetic position of the hypnosqualean clade as highly derived Squalomorphii, the diagonal tooth arrangement in Hypnosqualea was retrieved as a secondary gain in this group and was discussed as a synapomorphy for them [47]. In our analysis, this state, conversely, is retrieved as the plesiomorphic condition for elasmobranchs, with two independent gains in Hexanchiformes and Squaliformes.

(6) *Metapterygial condyle present*: As discussed in the previous section, the traditional view is that elasmobranchs have a single condyle on the scapulocoracoid supporting the

pectoral basals with only Squatiniformes having a separate condyle for the metapterygium and Pristiophoriformes and Batomorphii having a third condyle for the propterygium [44–46,48,77,78]. However, recent morphological studies on the pectoral girdle of elasmobranchs revealed a much more complex organization of the articulation surface of the scapulocoracoid in many groups [133,137]. The presence of a separate condyle for the metapterygium was observed in Squatiniformes, but also in many orectolobiform sharks (*Ginglymostoma*, *Stegostoma*, and *Rhincodon*) [133]. Several lamniform as well as squaliform sharks of the family Etmopteridae also were reported to have two separate condyles on the scapulocoracoid, one for the procondyle and one for the meso- and metacondyle [133,137]. Our character coding on the articulation of the pectoral radials followed these new observations and did not recover any unambiguous synapomorphies for Squatiniformes + *Protospinax* or a hypnosqualean clade in our analyses.

(7) *Condition of anterior pelvic basal*: The anterolateral orientation of the anterior pelvic basal (as opposed to the posterolateral orientation in most elasmobranchs) was reported in Squatiniformes, Hexanchidae, and several Squaliformes of the family Etmopteridae (*Aculeola*, *Centroscyllium*, and *Trigonognathus*) [45,47,242]. In the disarticulated *Protospinax* specimen MCZ VPF-6394, the anterior pelvic basal is preserved on one side and points anterolaterally (de Carvalho and Maisey [47] fig 9C). Our revision of the available material for *Protospinax annectans* confirms the presence of this character state in more completely articulated specimens (MB 14-12-22-1; SNSB-BSPG 1963-I-19; UMN uncatalogued).

(8) *Plate-like supraneurals in abdominal region*: This character was reported in Squatiniformes, Pristiophoriformes, and several batomorph species (Rhinopristiformes and Torpediniformes) [45,232]. Expanded supraneurals are also present anterior to the second dorsal fin in *Squalus* [45,232], which, however, do not extend to the abdominal region. *Protospinax annectans* exhibits expanded supraneurals along the abdominal region. Our analysis suggests that this character is highly homoplastic and may represent an adaptation to a benthic lifestyle. However, it must be noted that benthic galeomorphs (even dorsally flattened species such as sharks of the genus *Orectolobus*) do not possess plate-like supraneurals.

(9) *Dorsal basal plates anchored to the vertebral column*: Large expanded basal plates support the dorsal fins in Squaliformes, Squatiniformes, Pristiophoriformes, Heterodontiformes, and Batomorphii [45]. The number of plates and their association with the vertebral column can differ within these groups. Squaliformes and Heterodontiformes usually have a single basal plate, which supports the dorsal fin spine, but the basal plate is also retained in squaliform sharks which lack dorsal fin spines. Whereas the basal plate in Heterodontiformes contacts the vertebral column along its whole ventral margin (Shirai [45] plate 55G), the basal plate in Squaliformes is not closely associated with the vertebral column with the exception of *Squalus acanthias* (Shirai [45] plate 54). In Pristiophoriformes and Squatiniformes, the dorsal fin is supported by two large and several smaller plates, which are in close proximity to the vertebral column (Shirai [45] plate 54E,F). De Carvalho and Maisey [47] recognized similarities between *Protospinax* and Squaliformes (single dorsal plate supporting a fin spine) but noted that the basal plate in *Protospinax* is in close proximity to the vertebral column along its whole ventral margin, such as in Pristiophoriformes and Squatiniformes. Our reexamination revealed that the basal plates in most *Protospinax* specimens are poorly preserved, with the exceptions for specimens JME-SOS 3386 (Figure S4) and in MCZ VPF-278 (see de Carvalho and Maisey [47] figs. 3 and 8), in which they are very well-preserved. Despite this shared feature, there are differences in the dorsal fin endoskeleton between *Protospinax* and the other two clades, i.e., the presence of dorsal fin spines and the presence of a single dorsal basal plate in *Protospinax* (as opposed to two large dorsal basals without fin spines in Squatiniformes and Pristiophoriformes). To better cover the variation observed in the configuration of the basal plates in these groups, a new character was added to the data matrix (char. 155: number of basal plates (if the endoskeleton is composed of triangular or rectangular basal cartilage)).

Although our analysis included new morphological characters, changed the previous coding slightly (as discussed above), and was constrained by a molecular backbone, the resulting phylogenetic position of *Protospinax annectans* changed mainly in regards to the Batomorphii, whereas a close relationship between *Protospinax*, Squatiniformes, and Pristiophoriformes is supported (as proposed in de Carvalho and Maisey [47]). A sister group relationship between Squatiniformes and Pristiophoriformes is supported by many molecular phylogenetic analyses [37,39–41,43,203,243], although this scenario is challenged by a few other studies [42,62]. Although we recovered *Protospinax annectans* at the base of a clade comprising Squatiniformes and Pristiophoriformes (like in de Carvalho and Maisey [47]) in our majority-rule consensus tree (Figure 10), the strict consensus tree revealed that this group is not supported by any unambiguous synapomorphies and that the phylogenetic interrelations of *Protospinax* remained unresolved within Squalomorphii (Figure 9). More importantly, our revision of previously proposed characters defining a group comprising *Protospinax*, Squatiniformes, and Pristiophoriformes revealed that all of them were either ambiguous or homoplastic. It cannot be ruled out that these homoplastic characters uniting *Protospinax* with Squatiniformes and Pristiophoriformes are simply the result of the shared benthic ecomorphology (for a paleoenvironmental reconstruction of *Protospinax annectans*, see Figure 11). Therefore, we regard the proposed phylogenetic position of *Protospinax annectans* at the base of a clade comprising *Protospinax*, Squatiniformes, and Pristiophoriformes as provisional, pending further investigation.

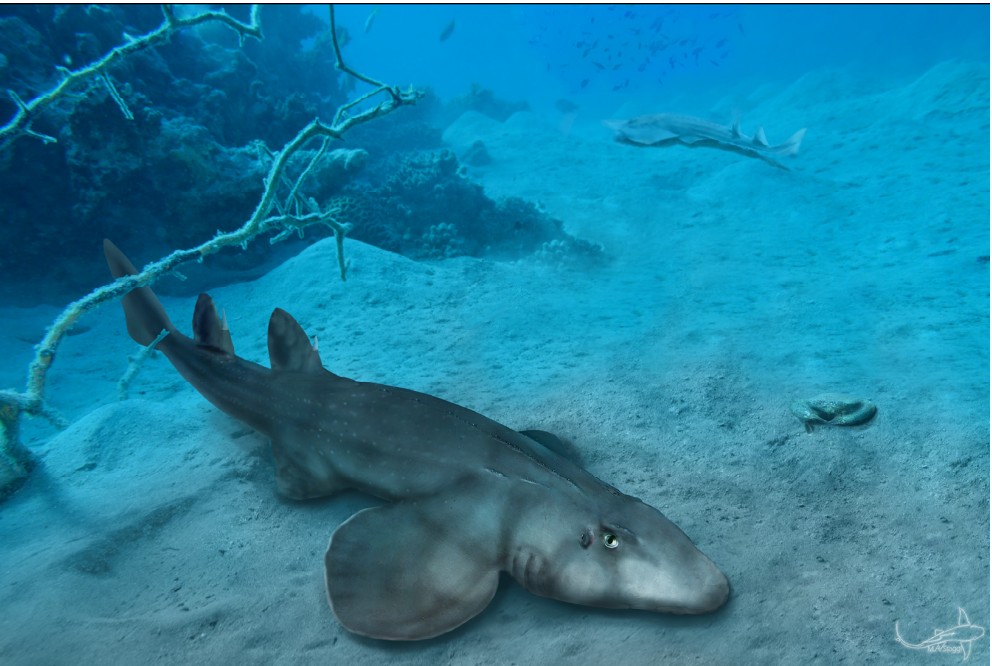

**Figure 11.** Environmental reconstruction of the Tithonian (Late Jurassic) Solnhofen Archipelago, showing *Protospinax annectans* in association with the Late Jurassic ray *Asterodermus platypterus*.

### 4.3.3. *Protospinax* as A Squaliform Shark as Proposed by Villalobos-Segura et al. [59]

Contrary to de Carvalho and Maisey's [47] and our results, Villalobos-Segura et al. [59] did not recover a clade comprising *Protospinax annectans*, Squatiniformes, and Pristiophoriformes, but proposed a closer relationship between *Protospinax* and Squaliformes. According to that hypothesis, *Protospinax* would be the oldest known squaliform shark and possibly represent a stem group member for this group. A closer relationship between *Protospinax* and Squaliformes, however, was only supported by a single character, the presence of two fin spines [59]. Although it is true that within extant Squalomorphii, Squaliformes is the only group exhibiting this feature, two dorsal fin spines are widespread among Chondrichthyes, and a secondary loss of this feature could have occurred in Squatiniformes

and Pristiophoriformes. However, it would be premature to dismiss the possibility of a closer relationship between *Protospinax annectans* and Squaliformes, because a look at the synapomorphies holding squaliform sharks as a monophyletic group in our analysis reveals that they were mainly defined by dental characters and only a single skeletal character: the presence of the subnasal fenestra (Figure 9). Although it is widely accepted that shark teeth can bear a phylogenetic signal [60,61,97,99,244]), it was also shown that they can cause significant noise in phylogenetic analyses [99]. This does not come as a surprise, as shark teeth are highly functional units and closely resembling morphologies can evolve independently as a result of adapting to similar food resources. In fact, if the scenario holds true that *Protospinax* was a basal squaliform shark, the lack of skeletal characters uniting this group and the dramatic change in dentition (possibly due to a shift in diet) would make it impossible to resolve this relationship in our analyses. It therefore is essential to find and define more skeletal characters for this group, in order to be able to test the hypothesis that *Protospinax* was a basal squaliform shark.

## 5. Conclusions

The present study utilized morphological and molecular evidence to revise the phylogenetic interrelationships of the enigmatic Late Jurassic elasmobranch *Protospinax annectans*. The molecular backbone constraint applied on the morphological data set prevented the resurrection of the Hypnosqualean hypothesis and allowed us to test the phylogenetic position of this key taxon within a molecular framework that is also supported by the fossil record. Furthermore, it allowed us to test the morphological support of certain phylogenetic units.

A sister relationship between sharks (Squalomorphii and Galeomorphii) and rays (Batomorphii) as proposed by molecular evidence was only weakly supported by a single character, the three-layered enameloid. Whereas the three super orders were well-supported by molecular and morphological data, the relationships within the Batomorphii and Squalomorphii were far from settled. While batomorphs show generally high inconsistencies between different phylogenetic studies, the main problem within squalomorphs is the taxon *Echinorhinus* and the lack of characters defining the order Squaliformes.

Our analysis supports a clade comprising *Protospinax*, Squatiniformes, and Pristiophoriformes with *Protospinax* at its base in our majority consensus tree. However, the strict consensus tree revealed that no unambiguous synapomorphies are supporting this group and that the phylogenetic position of *Protospinax* within Squalomorphii remains ambiguous. Due to the lack of skeletal characters defining squaliform sharks and their highly derived dental characters, an alternative scenario in which *Protospinax* is a basal squaliform shark or sister group to a clade comprising Squaliformes, Squatiniformes, and Pristiophoriformes cannot be dismissed with certainty either. For the moment, the phylogenetic position of *Protospinax* is best regarded as tentative and its use as a calibration fossil for the divergence time of modern sharks and rays is highly questionable until its phylogenetic position can be resolved.

The molecular backbone constraint applied to our analysis was proven to be a useful tool to incorporate fossil taxa into a molecular hypothesis and to pinpoint weaknesses in the morphological data. Future studies can incorporate further fossil taxa, e.g., the Late Jurassic shark *Palaeocarcharias stromeri*, which is hypothesized to have played an important role in the evolution of lamniform sharks but is similarly controversial when it comes to its phylogenetic relations [77,107,245,246]. Ultimately, this approach has the potential to augment our understanding about the evolution of morphological characters and the evolutionary history of elasmobranchs further by combining morphological, molecular, and paleontological evidence within a standardized framework.

**Supplementary Materials:** The following supporting information can be downloaded at: https://www.mdpi.com/article/10.3390/d15030311/s1, Supplementary Material S1: Character matrix (nexus file) used in this study; Supplementary Material S2: Character list used in this study; Supplementary Material S3: TNT scripts; Supplementary Material S4: PAUP scripts; Supplementary

Material S5: Logfile TNT parsimony analysis with conservative molecular backbone constraint; Supplementary Material S6: Logfile TNT parsimony analysis with fully enforced molecular backbone constraint; Supplementary Material S7: Logfile TNT unconstrained parsimony analysis; Supplementary Material S8: Logfile PAUP unconstrained parsimony analysis; Supplementary Material S9: Logfile PAUP maximum-likelihood analysis; Figure S1: Molecular backbone constraint tree and posterior probability values generated from the dataset of Stein et al. [13]; Figure S2: Phylogenetic trees obtained from two different analyses: (A) strict consensus tree from an unconstrained parsimony analysis computed in PAUP and TNT; (B) single tree generated by the (unconstrained) maximum likelihood analysis computed in PAUP; Figure S3: Majority-rule consensus tree (A) and strict consensus tree (B) from a constrained parsimony analysis with fully enforced molecular backbone constraints; Figure S4: *Protospinax annectans* Woodward, 1918, JME-SOS 3386, from the lower Tithonian of the Solnhofen Archipelago, Bavaria, Germany. (A) whole specimen under normal light; (B) close-up view of the pelvic region; (C) close-up view of the mandibular arches; Figure S5: *Protospinax annectans* Woodward, 1918, MB 14-12-22-1, from Blumenberg, Eichstätt (Solnhofen Archipelago, lower Tithonian), Bavaria, Germany. (A) whole specimen under normal light; (B) close-up view of the pelvic region; (C) close-up view of the pelvic region with the skeletal elements of the clasper highlighted; Table S1: Vertebral measurements of PBP-SOL-8007. References [13,45–48,52,58–61,77–84,86,87,90,95–112,119,122–124,127,132,138,194,227,232,233] are cited in the supplementary materials.

**Author Contributions:** Conceptualization, P.L.J. and J.K.; methodology, P.L.J. and E.V.-S.; validation, P.L.J.; formal analysis, P.L.J.; investigation, P.L.J., E.V.-S., J.T., A.B. and J.K.; resources, R.K., S.K., F.L., B.P., J.G.M., G.J.P.N. and J.K.; data curation, F.L., B.P., G.J.P.N. and J.K.; writing—original draft preparation, P.L.J.; writing—review and editing, P.L.J., E.V.-S., J.T., A.B., M.A.S., S.S., J.G.M. and J.K.; visualization, P.L.J., J.T., M.A.S. and S.S.; supervision, J.K.; project administration, J.K.; funding acquisition, J.K. All authors have read and agreed to the published version of the manuscript.

**Funding:** This research was funded in whole by the Austrian Science Fund (FWF) [P 33820] and [P 35357]. For the purpose of open access, the author has applied a CC BY public copyright licence to any Author Accepted Manuscript version arising from this submission.

**Institutional Review Board Statement:** Not applicable.

**Informed Consent Statement:** Not applicable.

**Data Availability Statement:** All data used by the authors for the analysis are available in the Supplementary Materials.

**Acknowledgments:** We want to thank all curators and collection managers who provided us access to the examined material, including Amy Henrici (Carnegie Museum of Natural History, Pittsburgh), Margot Gerhäußer (Fossilien und Steindruck Museum, Gunzenhausen), Martina Kölbl-Ebert and Christina Ifrim (Jura-Museum Eichstätt), Georg Bergér (Museum Bergér, Eichstätt), Alan Pradel (Muséum National d'Histoire Naturelle, Paris), Emma Bernard and Zerina Johanson (Natural History Museum, London), Oliver Rauhut (Bayerische Staatssammlung für Paläontologie und Geologie, Munich), Sebastian Neiderhell (Urweltmuseum Neiderhell, Raubling), and Hans-Dieter Sues (Smithsonian National Museum of Natural History, Washington DC). Hans-Dieter Sues is also thanked for discussions about the International Code of Zoological Nomenclature (ICZN) and how to determine the date of publication. Special thanks go to Jürgen Pollerspöck (Shark-References database) for discussions about morphological characters in squaliform sharks and for assisting with literature essential for this study. We thank Brian Metscher (University of Vienna) for preparing the CT data for specimen EMRG-Chond-T-80. Faviel A. López-Romero (University of Vienna) is thanked for his help in pruning the original tree of Stein et al. (2018), especially after the tool on the vertlife.org webpage for generating subsets stopped working. Claudia Klimpfinger (University of Vienna) is thanked for discussions on labial cartilages in sharks and rays. Last, but not least, we are thankful to two anonymous reviewers for their constructive feedback. Open Access Funding by the Austrian Science Fund (FWF).

**Conflicts of Interest:** The authors declare no conflict of interest. The funders had no role in the design of the study; in the collection, analyses, or interpretation of data; in the writing of the manuscript; or in the decision to publish the results.

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
