# Peer review of "Systematics and Phylogenetic Interrelationships of the Enigmatic Late Jurassic Shark Protospinax annectans Woodward, 1918 with Comments on the Shark–Ray Sister Group Relationship"

_diversity, doi:10.3390/d15030311_

Round 1

Reviewer 1 Report

Dear editor(s),

Dear authors,

It was my pleasure to read and review the manuscript of Jambura et al. entitled “Systematics and phylogenetic interrelationships of the enigmatic Late Jurassic shark Protospinax annectans Woodward, 1918 with comments on the shark-ray sister group relationship”.

The manuscript is well written, well structured, and prepared with much care, and this into every detail. The methodology is good and I have little to no general remarks. For smaller comments, corrections and suggestions, please check the attached annotated manuscript carefully.

Best regards

Author Response

Reviewer 1:

Dear editor(s),

Dear authors,

It was my pleasure to read and review the manuscript of Jambura et al. entitled “Systematics and phylogenetic interrelationships of the enigmatic Late Jurassic shark Protospinax annectans Woodward, 1918 with comments on the shark-ray sister group relationship”.

The manuscript is well written, well structured, and prepared with much care, and this into every detail. The methodology is good and I have little to no general remarks. For smaller comments, corrections and suggestions, please check the attached annotated manuscript carefully.

Best regards
Response: We are jubilant about the reviewer’s positive feedback and are thankful for the thorough review. We went through the annotated manuscript and revised our manuscript accordingly. We hope we were able to address each of the reviewer’s comments sufficiently.

Line 35: ?fossil calibration
Response: The keyword “calibration fossil” was intended as Protospinax annectans has been frequently used as one.

Line 57: Suggestion: add main reference Volta (1796) - Ittiolitologia Veronese.
Response: We added the reference as suggested by the reviewer.

Line 58: Suggestion: add locality Froidefontaine (France) that is quite similar to that of Rauenberg (Germany).

For main reference on sharks see Pharisat (1991) - La Paeoichthyofaune du Rupelien marin de Froidefontaine (Territoire de Belfort): Taxinomie et populations, genese du gisement, implications paleobiogeographiques. Annales scientifiques, Université Franche–Comté, Besançon, Géologie, 4, 13–97.

See also Pharisat (1991) - https://doi.org/10.1080/11250009809386811
Response: We added the locality and the reference as suggested by the reviewer.

Line 86-89: It reads like an update of a summary/list that was first presented by Maisey in 1976 [50]. But OK, I also see that you have Maisey in line 91. But make sure that you keep the Maisey reference close to it.
Response: It is true that a similar summary has been given by Maisey 1976 and later by de Carvalho & Maisey 1996. However, we feel there is nothing wrong with citing the primary literature instead of the secondary literature.

Line 94: Just adding this taxon name between brackets is very confusing for the reader. What do you mean: a) that both species are synonyms, b) that annectans was assigned to Belemnobatis, c) a combination of a-b? I guess you want to say 'b'? In that case be straight forward and say that Maisey re-allocated annectans to Belemnobatis.
Response: This sentence was rephrased as suggested by the reviewer to make it clearer that Protospinax annectans was reassigned to Belemnobatis annectans in the cited work:
“He concluded that the new specimen and the holotype belonged to the Batomorphii and were closely allied with the Late Jurassic batomorph Belemnobatis sismondae, leading to their assignment to Belemnobatis annectans, whereas the paratype was described as a new galeomorph shark species, Squalogaleus woodwardi [54].”

Line 94-98: This is confusing. I think that you want to say that annectans was re-allocated from Belemnobatis to Protospinax again, not? If Carvalho & Maisey really did synonymize Belemnobatis (annectans) with Protospinax (annectans), then Belemnobatis would have priority, and Protospinax is no longer a valid taxon. Please rephrase lines 94-98 very carefully
Response: That’s not really correct. It is true that we meant that the species “annectans” was re-allocated from the genus Belemnobatis to Protospinax. Nonetheless, this leads to Belemnobatis annectans Maisey, 1976 being a synonym of Protospinax annectans Woodward, 1918, as reflected by it being listed in the synonym list of Protospinax annectans in de Carvalho & Maisey 1996. Therefore, we feel that the original phrasing of our sentence was accurate.

Line 104: Suggestion: 'came from LATER ...' Otherwise the past tense is confusing, knowing that studies 56 and 57 were published after 43.
Response: The sentence was slightly rephrased to resolve this issue:
“Support for this hypothesis came from subsequent phylogenetic studies based on dental characters, which also proposed a close relationship between Protospinax, Squatiniformes, and Pristiophoriformes [60,61]”

Line 110: Here you come very close to an exact phrase used in the introduction of Maisey (1976) [ref 50, but still OK. Perhaps add ref 50 here again.
Response: Although the shared phrasing is mainly restricted to the word “allotted”, we took the concerns of the reviewer into account and slightly rephrased the sentence and added the reference as it was suggested.

Line 141: At first sight, this tooth is not listed in the material section. Does it originate from one of the holomorphic specimens, or not? Please specify.
Response: We rephrased this section to make it clear that the tooth used for micro-CT imaging was found isolated and not in association with any of the holomorphic specimens:
The tooth histology of Protospinax annectans  was examined non-destructively with X-ray computed tomography (CT) imaging using a Xradia MicroXCT-system (Zeiss, Oberkochen, Germany) at the Department of Theoretical Biology (University of Vienna, Austria). The examined tooth (EMRG-Chond-T-80) was from the Solnhofen Archipelago but was found isolated and was not associated with any holomorphic specimen.

Line 153 & 645: Perhaps stress that in Supplementary Materials (Supplementary Table S1).
Response: Although the suggested abbreviation would be also our personal preference, we’re afraid that the MDPI Author Guidelines require us to leave it as it stands now.

Line 157: It helps the reader if you would start this sentence for instance as follows: 'Within the ingroup,....'.
Response: The sentence was rephrased accordingly.

Line 172 & 774: Sometimes you use 'char.', sometimes you use '#' I think. Please check, and if so, please make consistent throughout entire ms.
Response: “#” was changed to char. throughout the whole manuscript.

Line 232: It seems that I can't find this acronym in the list of institutional abbreviations?
Response: The acronym was slightly misspelled (an old version was written on the specimen) and was amended accordingly throughout the manuscript.

Line 265: Suggestion to delete this info here. This info is also mentioned at the end of this section 2.4.
Response: Amended as suggested.

Line 288: Add JM-SOS
Response: Done.

Line 321: Perhaps add '(monogeneric family)' to be straightforward
Response: Done.

Line 327: Cappetta (2006, 2012) also listed P. muftius. Perhaps explain why this species is not listed here anymore.
Response: The species “P. muftius” and an explanation why we do not consider it a valid species has been added.

Line 368: Remove bold.
Response: Amended.

Line 374: Better: originate.
Response: Although the word “originate” would be correct, some readers might feel that we want to express that the species Protospinax annectans originated in this area. To avoid this confusion, we left the original phrasing here.

Line 425: Numbering is not correct. Please correct throughout ms.
Response: We really appreciate the thoroughness of the reviewer and amended the numbering accordingly.

Line 438 & 1995: I is a pity that you only list measurements of PBP-SOL-8007, whereas more specimens have been included in this study. Adding morphometrics for the other material is even more of interest in view of the fact that some specimens are a bit different from each other in appearance.
Response: Only 3 specimens were well enough preserved to take these measurements. We added them and the holotype in the revised version of the manuscript to show the intraspecific variation in Protospinax annectans.

Line 446: You always repeat the word 'Figure'. Better to combine as Figure 3-4 etc... Please check throughout entire ms.
Response: This formatting does not stem from our own preferences but is imposed by the guidelines of the journal.

Line 499: I understand that you want to keep 'dentition' next to the section 'mandibular arch'. However, by doing so, you are mixing endoskeletal and exoskeletal elements in this chronology. Perhaps consider to keep 'dentition' 'dorsal fin spines' and 'dermal denticles' together.
Response: As indicated by the reviewer, we were simply following the topology of the structures in question. We feel there is nothing wrong with either approach but would prefer to keep our ordering, because moving the teeth at the end of the description would require removing them from Figure 4 (where they are depicted together with three panels illustrating the skull) and rearrange Figure 4 or leave a plank panel.

Line 528: ? delete one blank line here
Response: Amended.

Line 702: Even if repeated over and over again in each of the sections, I would suggest to mention the specimen again (I guess PBP-SOL-8007).
Response: Amended.

Line 720: Enter a space here. Delete semicolon.
Response: Amended.

Line 733: Why not for hydro-dynamics as well? There is some controversy about this.  Some reference(s) would be of interest here. See e.g. hotly debated recent paper by Macias-Cuyare & Oddone, 2022 - 'abrasian stenght'. https://onlinelibrary.wiley.com/doi/10.1002/jmor.21493
Response: The undifferentiated morphology, lacking keels and ridges, indicate that these denticles on the snout had no hydrodynamic function, but solely served protection (abrasion resistance). This is also indicated by the suggested paper, which we included to strengthen our argument here.

Line 736: 8D-F
Response: Amended.

Line 750: Check numbering (see comment above).
Response: The wrong numbering Line 425 only affected the sub header 2 (3.1.X), but not the sub-header (3.X) here.

Line 846: I would suggest to highlight the so-called Hypnosqualea clade in Figure S2 (A and B) by marking it with a red rectangle or so, and add a note in the caption of S2.
Response: Figure S2 was amended and highlights the “Hypnosqualea” clade now.

Line 1096: And what about Echinorhiniformes? I've not checked coding in your matrix... Also, Underwood et al. (2016) were not straight forward in the interpretation of Squatina.
Response: Teeth of Echinorhiniformes was also coded to be linearly arranged along the jaw margin without an imbrication. This was amended in the text accordingly. About Squatina: it is true that adult Squatina superficially exemplifies a single file dentition, however, Smith et al. 2018 (also cited in this section together with Underwood et al. 2016) showed that the tooth initiation of neighbouring tooth files exhibit an alternate timing (as in other sharks with diagonally arranged teeth) and thus was coded as having diagonally arranged teeth.

Line 1145-1148: Suggestion: do not use capital letters for figs and/or plates from other papers.
Response: Amended.

Line 1193: Perhaps you can stress the value of this assumptions by adding/referring to the same hypothesis that has been put forward in other taxonomic groups. A good example might be Synechodontiformes/ Palaeospinacidae. See for instance Klug (2010) and short discussion in Mollen & Hovestadt (2018).

https://onlinelibrary.wiley.com/doi/abs/10.1111/j.1463-6409.2009.00399.x

https://sciencepress.mnhn.fr/en/periodiques/geodiversitas/40/25

Response: we appreciate this advice, however, the monophyly of the order Synechodontiformes is still controversially debated (even among the authors of this paper) and we thus do not feel that adding this controversy would strengthen our argumentation here significantly. 

Line 1198: Although
Response: Amended.

Line 1565: These Herman papers are often not cited correctly. In fact, in yellow is the name of the series with Stehmann the sole editor. So correct citation is Herman .... Part A.... in Contributions ... (Stehmann, ed.). Check front pages of all Herman papers.
Response: We admit that citing the work of Herman et al. in Bulletin de L'Institut Royal des Sciences Naturelles de Belgique, Biologie is challenging and many different versions are out there. However, the reference type suggested by the reviewer would imply that the work of Herman was part of a book, which is not the case. Bulletin de L'Institut Royal des Sciences Naturelles de Belgique, Biologie is a journal and we chose to use the citation form for such. Also, if self-citations of Herman et al. are reviewed it is apparent that “Contributions to the Study of the Comparative Morphology of Teeth and Other Relevant Ichthyodorulites in Living Supraspecific Taxa of Chondrichtyan Fishes” is not treated as a separate or subordinate publication, but is part of the title (as applied by us). The inclusion of an editor in a journal article is uncommon and we leave it to the technical editor here to decide whether or not this should be added.

Line 1567: Sometimes in full, sometimes abbreviated. Please make consistent.
Response: Amended.

Line 1908: General comment on figures. Now they are all inserted at the end of the manuscript. Suggestion to integrate them better in the manuscript itself.
Response: Some of the figures are quite large and inserting them to the proper section wasn't always easy, which is why we wanted to leave it to the expertise of the technical editor to put them at a proper position.

Although the comments and suggestions made by the reviewers were indicated as minor, we feel they substantially improved the quality of the manuscript. For that, I would like to express my gratitude on behalf of all my co-authors.

Sincerely,

Patrick L. Jambura

Reviewer 2 Report

Dear authors:

The paper titled “ Systematics and Phylogenetic Interrrelationships of the Enigmatic Late Jurassic Shark Protospinax annectans Woodward, 1918 with Comments on the Shark-Ray Sister Group Relantiosnship” deal with the very interesting problem of the placement of controversial fossil Protospinax annectans in the chondrichthyan phylogeny using morphological characters under a molecular backbone constrain. Not only that, but through a detail review of those morphological characters the authors reach the conclusion, to which I agree, that a revision of those traits is needed to obtain more unambiguous synapomorphies to define the major groups within Elasmobranchii (Batomorphii, Squalomorphii and Galeomorphii) and their validity for phylogenetical studies.

In my opinion, it is a work of great scientific interest and provides data that will be really useful for other researchers. Despite all the expose above, I have some minor comments and /or suggestions that I would like to ask the authors to consider:

- My main issue is the election of the molecular backbone constraint tree of Stein et al. (2018). In this work, one of the fossil taxa they use to time calibrate their molecular tree is Protospinax annectans. Not only Stein et al. place Protospinax annectans as a stem of the superorder Squalomorphii, a clade including Hexanchiformes, Pristiophoriformes, Squaliformes and Squatiniformes; but they regard this taxon as one of the three exceptions of fossil that show the diagnostic apomorphies needed to be used in phylogenetical analysis. The authors stated that: “The molecular backbone constraint applied on the morphological data set did prevent the resurrection of the Hypnosqualean hypothesis and allowed us to test the phylogenetic position of this key-taxon within a molecular framework that is also supported by the fossil record” (lines 1211-1214). Doesn’t this seem to be some circular reasoning? How can the authors know that this doesn’t bias their results, particularly since the placement of Protospinax annectans as a fossil calibrate in the molecular framework seems to be the same as their result? Maybe you should include a statement explaining why you chose to use that particular molecular framework as a constraint, although it was calibrated with the same fossil taxon that they want to position in the phylogeny.

- Page 8, lines 378-401: The section 3.1.1 Nomenclatural notes , although informative doesn’t add any relevant data to the topic discussed in the manuscript. I would suggest deleting it.

- Page 9 line 425: Change 3.1.2. Description to 3.1.3. The subtitle 3.1.2 has been already use for the revised diagnosis.

- Page 9 line 427: Change Figure 2A to Figure 1A

- Page 15 line 736: Figure D-F, change to Figure 8 D-F

- Page 23 line 119: Change Al1hough to Although.

Author Response

Reviewer 2: Dear authors:
The paper titled “ Systematics and Phylogenetic Interrrelationships of the Enigmatic Late Jurassic Shark Protospinax annectans Woodward, 1918 with Comments on the Shark-Ray Sister Group Relantiosnship” deal with the very interesting problem of the placement of controversial fossil Protospinax annectans in the chondrichthyan phylogeny using morphological characters under a molecular backbone constrain. Not only that, but through a detail review of those morphological characters the authors reach the conclusion, to which I agree, that a revision of those traits is needed to obtain more unambiguous synapomorphies to define the major groups within Elasmobranchii (Batomorphii, Squalomorphii and Galeomorphii) and their validity for phylogenetical studies.

In my opinion, it is a work of great scientific interest and provides data that will be really useful for other researchers. Despite all the expose above, I have some minor comments and /or suggestions that I would like to ask the authors to consider:
Response: Again, we want to thank the reviewer for taking their precious time to review and improve our manuscript. We are pleased with the overall positive response from the reviewer and hope to have addressed his/her/their response adequately.

- My main issue is the election of the molecular backbone constraint tree of Stein et al. (2018). In this work, one of the fossil taxa they use to time calibrate their molecular tree is Protospinax annectans. Not only Stein et al. place Protospinax annectans as a stem of the superorder Squalomorphii, a clade including Hexanchiformes, Pristiophoriformes, Squaliformes and Squatiniformes; but they regard this taxon as one of the three exceptions of fossil that show the diagnostic apomorphies needed to be used in phylogenetical analysis. The authors stated that: “The molecular backbone constraint applied on the morphological data set did prevent the resurrection of the Hypnosqualean hypothesis and allowed us to test the phylogenetic position of this key-taxon within a molecular framework that is also supported by the fossil record” (lines 1211-1214). Doesn’t this seem to be some circular reasoning? How can the authors know that this doesn’t bias their results, particularly since the placement of Protospinax annectans as a fossil calibrate in the molecular framework seems to be the same as their result? Maybe you should include a statement explaining why you chose to use that particular molecular framework as a constraint, although it was calibrated with the same fossil taxon that they want to position in the phylogeny.
Response: There seems to be a misunderstanding here. The main reason behind the molecular backbone was to test the phylogenetic placement of Protospinax annectans under the assumption that the molecular evidence was correct and the hypnosqualean hypothesis (generated from morphological data) can be dismissed. An additional line of evidence for the molecular hypothesis is the fossil record (as mentioned above). However, we don’t see the circular reasoning here, it was simply our working hypothesis (nonetheless, the alternative, the hypnosqualean hypothesis was also presented here and not completely left out).

We also see no circular reasoning in using the molecular backbone tree of Stein et al. 2018, because, although Protospinax was used as a calibration fossil (as a squalomorph stem-group member), its phylogenetic position hasn’t been tested in Stein et al. 2018 and no phylogenetic analysis has ever proposed that Protospinax was situated at the base of the Squalomorphii (neither does any of our analyses). Furthermore, no constraint was applied on the fossil taxa, allowing them to freely move within the constrained tree.

The reasoning behind using the molecular tree of Stein et al. 2018 was simply that it is one of the most up-to-date phylogenies for Chondrichthyes and, to the best of our knowledge, is the best taxon sampled tree to date, including all extant genera that we were able to collect morphological data for.

- Page 8, lines 378-401: The section 3.1.1 Nomenclatural notes , although informative doesn’t add any relevant data to the topic discussed in the manuscript. I would suggest deleting it.
Response: The aim of this study was twofold. First, it should be a phylogenetic revision of Protospinax and morphological characters used in phylogenetic analyses. Secondly, it should be a comprehensive systematic/taxonomic review of the species and we feel that the correct date for erecting the species is not irrelevant for a taxonomic work, especially in the light of the lacking consensus here for the last 100 years. We therefore hope that the reviewer and the editor can see the importance of this section and accept our decision to leave it in the manuscript.

- Page 9 line 425: Change 3.1.2. Description to 3.1.3. The subtitle 3.1.2 has been already use for the revised diagnosis.
Response: Amended.

- Page 9 line 427: Change Figure 2A to Figure 1A
Response: Amended.

- Page 15 line 736: Figure D-F, change to Figure 8 D-F
Response: Amended.

- Page 23 line 119: Change Al1hough to Although
Response: Amended.

Although the comments and suggestions made by the reviewers were indicated as minor, we feel they substantially improved the quality of the manuscript. For that, I would like to express my gratitude on behalf of all my co-authors.

Sincerely,

Patrick L. Jambura